# Impact of Transitioning to Treated Water on Diarrhea Reduction: A Cross-Sectional and Ecological Study in Southwestern Goiás, Brazil

**DOI:** 10.3390/ijerph22030436

**Published:** 2025-03-16

**Authors:** Laise Mazurek, Camila Botelho Miguel, Henrique Polizelli Pinto Neto, Eduardo Henrique Vieira Araujo, Melissa Carvalho Martins de Abreu, Jamil Miguel Neto, Glicélia Pereira Silva, Mariana Santos Cardoso, Siomar de Castro Soares, Aristóteles Góes-Neto, Carlo José Freire Oliveira, Wellington Francisco Rodrigues

**Affiliations:** 1Postgraduate Program in Tropical Medicine and Infectious Diseases, Federal University of the Triângulo Mineiro—UFTM, Uberaba 38025-180, MG, Brazil; laise@unifimes.edu.br (L.M.); camilabmiguel@hotmail.com (C.B.M.); siomar.soares@uftm.edu.br (S.d.C.S.); carlo.oliveira@uftm.edu.br (C.J.F.O.); 2Multidisciplinary Laboratory of Scientific Evidence, University Center of Mineiros—Unifimes, Mineiros 75833-130, GO, Brazil; henriqueneto1211@hotmail.com (H.P.P.N.); dramelissa@unifimes.edu.br (M.C.M.d.A.); jamil@unifimes.edu.br (J.M.N.); glicelia@unifimes.edu.br (G.P.S.); 3State Emergency Hospital of Goiânia Dr. Valdemiro Cruz, Goiânia 74820-300, GO, Brazil; edh.araujo@outlook.com; 4Department of Microbiology, Molecular and Computational Biology of Fungi Laboratory, Instituto de Ciências Biológicas, Universidade Federal de Minas Gerais, Belo Horizonte 31270-901, MG, Brazil; marianascardoso@yahoo.com.br (M.S.C.); arigoesneto@gmail.com (A.G.-N.)

**Keywords:** treated water, diarrhea prevention, waterborne pathogens, public health, southwestern Goiás

## Abstract

Access to safe drinking water is a global challenge, with significant disparities affecting public health and quality of life. This study evaluated the relationship between specific water parameters, public satisfaction with treated water, and diarrhea incidence in Southwestern Goiás, Brazil. The objectives were (1) to assess water parameters, including aluminum, iron, manganese, pH, hardness, fecal coliforms, and turbidity, in samples from springs, streams, and municipal supplies; (2) to evaluate residents’ satisfaction with municipal water and their reliance on untreated water sources; and (3) to analyze the impact of transitioning from untreated to treated water following the closure of a spring in 2017 on diarrheal diseases. A longitudinal observational study reviewed diarrhea cases from 2013 to 2019. Treated water met potability standards, while untreated springs showed significant contamination. Dissatisfaction with treated water correlated strongly with continued reliance on untreated springs (*p* < 0.05), increasing the diarrhea risk nearly ninefold (OR = 8.78; 95% CI = 4.37–18.29). The findings underscore the importance of transitioning to treated water for mitigating diarrheal diseases and enhancing public trust in water safety. This study provides a replicable and scalable approach for improving water sanitation management, addressing waterborne diseases, and supporting public health interventions in diverse global contexts.

## 1. Introduction

Access to safe drinking water remains unevenly distributed worldwide, with significant variations driven by geographic, economic, and social factors [1]. According to the 2023 United Nations World Water Development Report, approximately 3.6 billion people lack access to basic sanitation, and 2 billion are deprived of safe drinking water. These deficits, particularly in developing regions, are directly linked to negative health outcomes, with inadequate hygiene services contributing to 1.4 million deaths globally in 2019 [2]. These figures underscore the urgent need to improve water and sanitation services, especially in underserved populations.

In Brazil, the situation mirrors this global challenge. Data from the 2022 census by the Brazilian Institute of Geography and Statistics (IBGE) indicate that 24% of the population lacks proper sewage systems, and approximately 6 million people do not have access to potable water. Brazil’s geographic and economic diversity exacerbates disparities in water supply and sanitation services, particularly in rural and low-income areas [3]. Water supply sources vary widely: 82.9% of households rely on distribution networks, while others depend on deep wells (9%), shallow wells or phreatic sources (3.2%), and natural springs or fountains (1.9%) [3].

These disparities in water access pose significant public health risks, particularly for marginalized populations. Insufficient access to safe water contributes to the spread of infectious diseases such as diarrhea and increases the prevalence of chronic conditions that strain healthcare systems. Diarrhea remains a leading cause of morbidity and mortality among children under five, especially in regions lacking access to safe drinking water and proper sanitation [4,5]. The World Health Organization (WHO) estimates that over 500,000 children die each year from diarrheal diseases that could have been prevented through improved water, sanitation, and hygiene services [6]. Ensuring access to safe drinking water is one of the most effective strategies to reduce the burden of waterborne diseases globally.

A pivotal event in Mineiros, Goiás, occurred in 2017 with the closure of the Magnífica Spring, a significant untreated water source widely used by the community. The closure, prompted by concerns over contamination with fecal coliforms and other pathogens, forced a shift from reliance on untreated water to treated municipal water. This event provided a unique opportunity to evaluate how transitioning to treated water impacts public health outcomes, particularly diarrheal disease incidence. It also raised important questions about the role of consumer satisfaction with treated water in shaping public trust and usage behaviors.

Consumer satisfaction with municipal water services is increasingly recognized as a critical factor in public health and water management. Dissatisfaction with specific water parameters—such as taste, odor, or perceived safety—often leads consumers to seek alternative, untreated water sources, increasing their exposure to waterborne pathogens [7,8,9]. Studies show that perceptions of municipal water, even when meeting regulatory standards, can drive behaviors that undermine public health goals. For example, dissatisfaction with municipal water in Brazil and elsewhere correlates with increased reliance on untreated sources, particularly in rural and underserved areas [8]. In Goiás, research has demonstrated that consumer satisfaction is shaped not only by water parameters but also by the management and pricing of water services [7].

Moreover, trust in water systems is strongly linked to health outcomes. Populations that lack trust in the safety and reliability of treated water are more likely to bypass municipal supplies, increasing the risk of disease transmission. Understanding how consumer perceptions of municipal water intersect with health outcomes, such as diarrheal disease incidence, is essential for designing effective policies and improving water management strategies locally and globally.

This study investigated the relationships among public satisfaction with treated water, the use of untreated sources, and diarrheal disease incidence in Mineiros, Goiás, before and after the closure of the Magnífica Spring. Specifically, it evaluated the physicochemical and microbiological parameters of water from different sources (springs, streams, and municipal supplies), examined residents’ satisfaction with treated water, and analyzed how reliance on untreated sources influenced diarrheal diseases. While site-specific, this research provides insights that are applicable to similar contexts worldwide, offering a replicable framework for addressing water safety and public health challenges in underserved communities.

## 2. Materials and Methods

### 2.1. Ethical Aspects of the Research

All research procedures adhered to the ethical principles established by the Declaration of Helsinki and Brazilian National Health Council Resolution 466/12. The study was reviewed and approved by the Research Ethics Committee of the Centro Universitário de Santa Fé do Sul—UNIFUNEC under approval number 1.838.794/2016, with the Certificate of Presentation for Ethical Consideration number 59866616.2.0000.5428. Informed consent was obtained from all participants before their inclusion in the study.

### 2.2. Research Design

This study employed an ecological and repeated cross-sectional observational design conducted in the municipality of Mineiros, Goiás, Brazil, which has an estimated population of 70,081 inhabitants across an area of 9042.844 km^2^ [10]. The geographical context of the study is delineated in Figure 1, where Panel (A) situates the state of Goiás within the central–west region of Brazil, while Panel (B) provides a detailed representation of Mineiros within the state. The study aimed to investigate the relationships among public perceptions of specific water parameters, such as aluminum, iron, manganese, pH, hardness, fecal coliforms, and turbidity; the use of untreated water sources; and the incidence of diarrheal diseases over time. To achieve these goals, the study integrated water sample analyses, household surveys, and health record assessments, focusing on population-level associations and temporal changes.

The design was defined as repeated and cross-sectional because different households were recruited for each survey phase (2016 and 2020) to evaluate population-level trends in water usage and satisfaction.

No individual level follow-up was conducted, eliminating concerns about loss to follow-up, a common issue in longitudinal studies where participants drop out before the study is completed, potentially introducing bias. This study avoided such limitations by adopting a repeated cross-sectional design, in which different households were recruited at each phase (2016 and 2020), ensuring population-level comparisons over time without relying on the same individuals. For health outcomes, ecological analyses were conducted by examining aggregated diarrhea incidence data from 2013 to 2019 to explore population-level trends and associations with water source usage over time. The term “ecological” refers to the analysis of community-level data, rather than individual-level outcomes, which aligns with the focus on aggregate measures of water quality, public perception, and health impacts. Data collection occurred in two distinct phases: December 2016, representing the period before the closure of the Magnífica Spring, and March 2020, after its closure in 2017. The closure, which was prompted by contamination concerns, served as a critical event for analyzing the impacts of transitioning from untreated to treated water on public health outcomes. By structuring the surveys around these time points, the study was able to assess changes in satisfaction, water source usage, and health outcomes within the community.

Water samples were collected from four locations: Coqueiros Stream, Magnífica Spring, Perpétua Spring, and residential homes. Untreated sources such as the Magnífica and Perpétua Springs were commonly used as alternatives to municipal water before the spring’s closure, particularly by residents dissatisfied with treated municipal water. These natural water sources, although accessible, often presented risks due to contamination. In contrast, the municipal water supply, primarily sourced from Coqueiros Stream and treated locally, consistently met Brazilian potability standards through filtration, disinfection, and pH adjustment. To ensure comprehensive assessments, water samples were collected at three distinct geographic points along each natural source, with 20 min intervals between collections, enabling evaluations of both spatial and temporal variability in the physicochemical and microbiological parameters.

Households were selected using a random block sampling strategy to ensure geographic and socio-demographic representativeness across the municipality. Blocks were defined as clusters of residences within specific neighborhoods, ranging from central to peripheral areas. Only residents with at least five years of residency in Mineiros were included to capture long-term perceptions of water sources and health impacts. Surveys focused on evaluating satisfaction with municipal water, reliance on untreated sources, and demographic variables such as household size, age range, education level, and employment status of the household head. Data collected during each phase were compared to assess trends in satisfaction and water usage before and after the closure of the Magnífica Spring.

The study was adequately powered, with a sample size of 383 households and a post hoc power calculation indicating an alpha level of 5% and a power of 80%, allowing detection of a minimum effect size of 0.3 (Cohen’s d). This ensured robust comparisons between untreated and treated water usage and meaningful associations between water quality perceptions and public health outcomes.

The study pursued three primary objectives: to assess the physicochemical and microbiological quality of water from different sources, including springs, streams, and the municipal supply; to evaluate residents’ satisfaction with the municipal water supply and their reliance on alternative water sources; and to analyze how the use of untreated water sources influenced the incidence of diarrheal diseases, particularly after the closure of the Magnífica Spring in 2017. By employing a mixed-methods approach combining ecological analyses and repeated cross-sectional surveys, this study provides a replicable framework for understanding the interplay among water quality, public perceptions, and health outcomes in communities transitioning to safer water sources.

### 2.3. Data Collection Procedures

For household water samples, tap outlets were aseptically cleaned using 70% alcohol, followed by a one-minute flush of water before collection. Each sample, with a volume of 1 L, was collected in sterile bottles, properly labeled, and stored in expanded polystyrene (EPS) boxes with ice packs to maintain the samples’ integrity during transport to the laboratory [11].

Sampling took place between 7:00 a.m. and 11:00 a.m. on designated collection days to minimize temporal variability across all locations. Quality control measures, including the use of sterile containers and immediate refrigeration, were consistently applied during transport [12]. Upon arrival at the laboratory, all samples were processed within four hours, following recognized standards for water sample preservation and analysis to ensure accuracy and reliability [11].

To capture both temporal and spatial variability, samples from each water source (Coqueiros stream, Magnífica Spring, and Perpétua Spring) were collected in triplicate. For each source, samples were taken from three distinct geographic points, with a 20 min interval between collections at each point, resulting in a comprehensive dataset that reflects potential variations across sampling locations. This triplicate sampling approach was specifically chosen to enhance analytical robustness, ensuring that any observed trends represent the general water quality across the sources rather than isolated measurements.

In the statistical analysis, including tests of correlation between turbidity and fecal coliform levels, the entire triplicate dataset was used. This method minimized the influence of any single sample, ensuring that correlations reflect trends across multiple observations rather than isolated outliers. Thus, it strengthened the reliability of the associations observed between water quality parameters, offering a thorough representation of each source’s water quality.

Household surveys were also conducted during these collection phases to assess residents’ satisfaction with the municipal water supply and their reliance on untreated sources, such as the springs. These surveys provided individual-level data on water usage habits, which were subsequently cross-referenced with population-level epidemiological data on diarrheal diseases from the Municipal Epidemiological Surveillance Department, covering the period from 2013 to 2019. This integrative approach allowed for a comprehensive analysis of the interrelations between water quality perceptions and public health outcomes.

### 2.4. Temporal Marker for Disease Incidence

A significant event in the study was the closure of the Magnífica Spring in 2017, which served as a temporal marker to compare water usage patterns and diarrheal disease incidence before and after the closure. This spring had been a widely utilized source of untreated water by the local population. Due to concerns about contamination, public access to the spring was officially terminated, compelling residents to transition to treated water provided by the municipal supply. This shift marked a decrease in reliance on untreated water sources, which was hypothesized to correlate with a reduction in diarrheal disease incidence in the municipality. Data collected after the spring’s closure (2017–2020) reflect this change in water consumption behavior and its impact on public health outcomes. The inclusion of both treated municipal water and untreated sources in the water samples prior to the closure allowed for a comprehensive evaluation of the health implications associated with water quality.

### 2.5. On-Site Measurements

The temperature and pH of the water were measured on site at the time of collection. A certified digital thermometer (Fisher Scientific, model 15–077-8, ±0.05 °C; Waltham, MA, USA) was used to record the water temperature. The pH was measured using a portable digital pH meter (Kasvi, model K39-0014PA^®^; São José dos Pinhais, PF, Brazil).

### 2.6. Laboratory Analyses

Laboratory analyses to detect and quantify iron, zinc, manganese, aluminum, turbidity, and fecal coliforms were conducted at Labfert Analyses in Uberaba, Minas Gerais, Brazil, following the Standard Methods for the Examination of Water and Wastewater, 22nd edition [13]. To ensure analytical reliability, all water samples were collected in sterile polyethylene bottles, pre-washed with ultrapure deionized water, and acidified with nitric acid (5% *v*/*v*) to prevent metal adsorption onto container surfaces. The samples were then stored at 4 °C in refrigerated transport units and analyzed within 48 h.

Prior to metal quantification, samples were filtered through 0.45 µm membrane filters to remove suspended solids. Acid digestion was performed according to Method 3030B, using nitric acid (HNO_3_) in Teflon or borosilicate glass digestion vessels, heated to 95 °C for 30 min, ensuring complete solubilization of the metal ions. For microbiological analyses, samples were maintained at 4 °C and processed within six hours under sterile conditions to prevent external contamination.

The quantification of iron, zinc, and manganese was performed using flame atomic absorption spectrometry (FAAS) with a Varian SpectrAA 220 spectrophotometer (Varian, Palo Alto, CA, USA), following Method 3111B. Aluminum was analyzed using graphite furnace atomic absorption spectrometry (GF-AAS) with a PerkinElmer PinAAcle 900T spectrophotometer (Shelton, CT, USA), following Method 3113B. Calibration for metal analyses was conducted using certified single-element and multi-element standard solutions (Merck^®^, 1000 mg/L), diluted in ultrapure deionized water to 0.05–1.00 mg/L for Fe, Zn, and Mn, and 0.01–0.50 mg/L for Al. Calibration curves were constructed using five concentration points, with blank samples, duplicate analyses, and internal standards incorporated to verify linearity and analytical precision. Instrument drift was monitored every 10 samples, and adjustments were made as needed to maintain analytical stability. The limit of detection (LOD) and limit of quantification (LOQ) for each element were as follows: Fe (LOD = 0.01 mg/L, LOQ = 0.03 mg/L), Zn (LOD = 0.005 mg/L, LOQ = 0.015 mg/L), Mn (LOD = 0.002 mg/L, LOQ = 0.007 mg/L), and Al (LOD = 0.001 mg/L, LOQ = 0.003 mg/L).

Turbidity was measured using a Hach 2100Q portable turbidimeter (Ames, IA, USA), following the nephelometric method (2130B), with calibration performed using formazin polymer standards (0.1, 1.0, 10, and 100 NTU). The LOD and LOQ for turbidity were 0.02 NTU and 0.05 NTU, respectively. Water hardness was determined using the EDTA titrimetric method (2340C), with calcium carbonate reference standards ranging from 50 to 500 mg/L, with an LOD of 5 mg/L and an LOQ of 15 mg/L.

The detection and quantification of fecal coliforms were performed using the multiple tube fermentation technique (9221), estimating the most probable number (MPN) based on probabilistic formulas. Samples were incubated at 44.5 °C for 24 h in lauryl tryptose broth (LTB) for the presumptive test, followed by a transfer to EC broth for confirmation of fecal coliforms. Positive controls (Escherichia coli ATCC^®^ 25922) and negative controls were included for method validation. The LOD for fecal coliforms was 1 MPN/100 mL, while the LOQ was 3 MPN/100 mL. Recovery control tests were periodically conducted to confirm the efficiency of viable organism detection, and the measurement uncertainty for microbial counts was ±8%.

Measurement uncertainty for metal quantification was calculated following ISO 11352 guidelines, employing Type A and Type B evaluation methods [14], considering factors such as instrumental variation, sample preparation errors, and matrix effects. The estimated expanded uncertainty values (95% confidence interval, k = 2) were ±5% for Fe, ±3% for Zn, ±6% for Mn, and ±4% for Al. The laboratory adhered to ISO 17025 accreditation standards [15], conducting all spectrometric measurements in triplicate to ensure analytical accuracy. Calibration curves were prepared using matrix-matched reference solutions, and stringent quality control measures were implemented, including duplicate sample analysis, spike recovery tests, and continuous instrument performance verification.

To contextualize our findings within international standards, we compared the measured concentrations with the regulatory thresholds established by the World Health Organization (WHO) and the European Union (EU). According to WHO guidelines, the maximum allowable concentrations are 0.3 mg/L for Fe, 0.05 mg/L for Mn, and 0.2 mg/L for Al [16]. The EU Drinking Water Directive (Directive 2020/2184) sets a maximum limit of 0.05 mg/L for Zn and 0.20 mg/L for Al [17]. Our results showed that all treated water samples complied with these guidelines, whereas the untreated sources (e.g., Magnífica Spring) frequently exceeded the recommended thresholds for Fe and Mn, reinforcing the necessity of effective water treatment and continuous monitoring.

### 2.7. Questionnaire on Water Use and Satisfaction

A structured questionnaire was administered in two phases (December 2016 and March 2020) to assess residents’ satisfaction with the municipal water supply and their reliance on untreated water sources, particularly after the closure of the Magnífica Spring in 2017. The questionnaire was concise, focusing on key aspects of water usage, quality perception, and satisfaction.

To differentiate between untreated spring water and the treated water provided by the municipal system, the questionnaire explicitly addressed these distinctions. Untreated spring water, such as the Magnífica Spring, was directly accessible to the public and widely used before its closure in 2017 due to contamination concerns. In contrast, the treated water is sourced, processed, and distributed by the municipality, meeting quality standards for public consumption. Survey questions were designed to ensure clarity in the responses by asking participants explicitly whether they consumed untreated water from springs or treated municipal water.

To gauge satisfaction with treated water, participants were asked questions such as, “Are you satisfied with the quality of treated water provided by the municipality?”, “Which aspects of the treated water do you consider most important for consumption (e.g., taste, smell, appearance)?”, and “Do you find the treated water supply reliable and sufficient for your daily needs?” Responses were categorized as satisfactory or unsatisfactory according to the frequency and consistency of concerns reported by participants.

This focused approach provided individual-level insights into residents’ preferences and behaviors related to water use, which were then cross-referenced with epidemiological records of diarrheal diseases from the Municipal Epidemiological Surveillance Department. By correlating public perceptions of water quality with health outcomes, this analysis enabled a robust understanding of the implications of shifting from untreated to treated water sources on public health.

### 2.8. Diarrhea Data Collection

Data on diarrhea cases from 2013 to 2019 were obtained from the Municipal Epidemiological Surveillance Department through a formal request, ensuring authorized access to anonymized health records in line with ethical guidelines. The dataset provided monthly case counts and the geographic distribution across the neighborhoods of Mineiros, enabling a comprehensive analysis of temporal and spatial trends in diarrheal disease incidence.

Initially, monthly case counts were analyzed to explore potential seasonal patterns, particularly considering the rainy and dry seasons typical of the region’s climate. However, monthly stratification revealed a bimodal distribution in diarrhea cases, with two main peaks per year, suggesting a cyclical trend in incidence. Given this observation, we structured our temporal analysis in six-month intervals or semesters to better capture these recurring incidence patterns and ensure analytical stability. This semester-based approach allowed us to compare trends across broader intervals, facilitating a more robust interpretation of the temporal data and controlling for shorter-term variability.

To allow for an accurate comparison between the two periods (pre- and post-spring water closure) and account for the differing durations, diarrhea incidence rates were calculated per 10,000 inhabitants annually. These incidence rates were presented as averages for each period, normalizing any discrepancies arising from the unequal number of years between the two intervals.

The health surveillance records included key variables, such as the date of diagnosis, neighborhood location, and patient demographics (age and sex), enabling a stratified analysis by demographic groups when relevant. To enhance data reliability, quality control measures were implemented to verify the accuracy and consistency of the data, including cross-referencing with health surveillance records and resolving any discrepancies identified during the data review.

The cross-referenced dataset was then integrated with household survey responses and water quality data, allowing the study to examine the associations among changes in water source usage, public perceptions of water quality, and patterns of diarrheal diseases within the community.

### 2.9. Data Analysis

The primary hypothesis of this study was that the reduction in the use of untreated spring water, driven by the closure of the Magnífica Spring in 2017, would correlate with a decrease in diarrhea cases across the community. This hypothesis reflects an ecological perspective, where population-level associations between untreated water usage and health outcomes are evaluated rather than individual case-level relationships. Consequently, the study is subject to the limitations inherent in ecological designs, including the risk of ecological bias and the potential for ecological fallacy.

Data were tabulated using Microsoft^®^ Excel and analyzed using SPSS software version 22.0. The Shapiro–Wilk test was used to assess the normality of the data. Spearman’s and Pearson’s correlation tests were applied to explore relationships between variables with abnormal and normal distributions, respectively. Chi-square tests were employed to evaluate the hypothesis that reduced use of spring water was associated with a decreased incidence of diarrhea on a community-wide scale. One-way ANOVA was used to compare water quality across the collection sites, followed by Dunn’s multiple comparison post-test. Logistic unadjusted regression analysis was performed to assess relationships with dichotomous outcomes. A significance level of 5% (*p* < 0.05) was adopted for all analyses.

## 3. Results

The physicochemical and microbiological analysis of water samples revealed significant variability in water quality across the different collection points, namely Coqueiros Stream, Magnífica Spring, Perpétua Spring, and residential taps. Analyzed parameters included iron, aluminum, zinc, manganese, hardness, turbidity, fecal coliforms, the correlation between turbidity and fecal coliforms, and pH (Figure 2).

Iron concentrations differed significantly among collection sites (*p* < 0.0001, Kruskal–Wallis test). Coqueiros Stream showed notably higher iron levels (mean = 0.3243 mg/L) compared with Magnífica Spring (mean = 0.0007 mg/L, *p* = 0.00059) and Perpétua Spring (mean = 0.0007 mg/L, *p* = 0.00026). In contrast, iron levels in residential water did not significantly differ from those in Coqueiros Stream (*p* = 0.518) (Figure 2A), indicating localized iron accumulation in natural sources. This may impact potability and pose health risks, especially for children, underscoring the need for regular monitoring of untreated sources.

Aluminum concentrations also varied significantly (*p* < 0.0001), with residential samples showing elevated levels (mean = 0.9175 mg/L) compared with Magnífica (mean = 0.0073 mg/L, *p* = 0.00213) and Perpétua Springs (mean = 0.0076 mg/L, *p* = 0.00050) (Figure 2B). In Coqueiros Stream, rgw aluminum levels emphasize the importance of monitoring and enhancing municipal water treatment to prevent contaminants from reaching the community’s treated supply. High aluminum exposure has been linked to neurotoxicity, stressing the need for improved water treatment protocols.

Zinc levels varied across sites (*p* = 0.00003), with Perpétua Spring showing the highest concentration (mean = 0.4825 mg/L) compared with Coqueiros Stream (mean = 0.0002 mg/L, *p* = 0.00086). No significant differences were observed between Coqueiros Stream and residential samples (*p* > 0.9999) (Figure 2C). Elevated zinc levels in natural sources may suggest environmental contamination, potentially from agricultural or industrial activities, requiring further investigation.

Manganese concentrations also showed variability (*p* = 0.00006), with Perpétua Spring exhibiting the highest concentration (mean = 0.6075 mg/L), significantly higher than Coqueiros Stream (mean = 0.003 mg/L, *p* = 0.017). Manganese levels in residential water (mean = 0.481 mg/L) were comparable with spring levels (Figure 2D). While manganese is necessary for health, chronic exposure to high levels may lead to neurotoxicity, making it important to monitor both natural and treated sources.

Water hardness was significantly higher in residential samples (mean = 125.2 mg/L, *p* < 0.0001) than in natural sources, which showed negligible hardness levels (mean = 0 mg/L for all) (Figure 2E). This indicates the presence of dissolved minerals like calcium and magnesium in municipal water, impacting taste and potentially causing scaling in pipes.

Turbidity levels varied significantly across sites (*p* = 0.0013), with Perpétua Spring having lower turbidity (mean = 1.393 NTU) compared with residential water (mean = 1.045 NTU), though the difference was not statistically significant (*p* = 0.09085) (Figure 2F). Although turbidity does not directly indicate microbial contamination, it is commonly used as a general water quality parameter. High turbidity can facilitate microbial growth by providing a medium for bacterial adhesion and protection from disinfection processes [18]. However, low turbidity does not guarantee microbial safety, as some pathogens, including viruses and free-floating bacteria, may be present in clear water. Therefore, turbidity should be interpreted cautiously and always complemented with direct microbiological analyses, such as the fecal coliform tests conducted in this study [19].

Fecal coliform levels were highest in natural sources, especially Coqueiros Stream (mean = 2300 MPN/100 mL), compared with residential water (mean = 1000 MPN/100 mL, *p* < 0.0001) (Figure 2G), indicating a higher risk of fecal contamination in untreated sources, which poses diarrheal disease risks to populations using these sources.

A significant correlation (r^2^ = 0.914, *p* < 0.0001) was found between turbidity and fecal coliforms, suggesting turbidity as an effective indicator of microbial contamination (Figure 2H). This correlation, based on data from triplicate samples across all collection points, strengthens turbidity’s utility in ongoing quality monitoring.

Lastly, pH varied significantly across sites (*p* < 0.0001). Magnífica Spring showed the most acidic levels (mean = 4.800), whereas residential water was within the potable range (mean = 6.070) (Figure 2I). The acidity of untreated sources may contribute to pipeline corrosion, highlighting the need for treatment before consumption.

Overall, variations across physicochemical and microbiological parameters underscore the need for continuous quality monitoring and effective treatment solutions to protect public health, particularly for communities that are reliant on untreated water sources.

An evaluation of the population’s satisfaction with the treated water provided by the municipality and its relationship with the use of alternative water sources was conducted among 383 respondents. Satisfaction levels were categorized into three groups: low, medium, and high (Figure 3). Among the respondents, only 27 individuals (7.1%) reported high satisfaction with municipal water, while the majority, 276 respondents (72%), expressed moderate satisfaction. A total of 80 respondents (20.9%) indicated low satisfaction with treated water.

This contingency table illustrates the association between satisfaction levels regarding water quality and the usage of untreated spring water in Mineiros, Goiás, Brazil, segmented by two periods: before and after the interdiction of the Magnífica Spring in 2017 due to contamination concerns (Table 1). Until 2016, a significant majority of individuals with low satisfaction (98.85%) relied on untreated spring water, compared with only 20.97% among those with high satisfaction levels. However, following the 2017 interdiction, there was a marked shift in water usage patterns; 100% of individuals with high satisfaction no longer used spring water, and the proportion of low-satisfaction individuals relying on spring water decreased to 83.91%. Chi-square analysis (χ^2^ = 161.1, *p* < 0.001) indicates significant changes in satisfaction and water source usage over time, reflecting the impact of restricted spring water access on community perceptions and health-related satisfaction (Table 1).

A logistic unadjusted regression analysis (Table 2) revealed a strong and significant correlation between dissatisfaction with municipal water and the use of alternative water sources, particularly untreated spring water. As satisfaction with treated water decreased, the likelihood of relying on spring water increased significantly (*p* < 0.05). Specifically, individuals with low satisfaction were over 75 times more likely to use spring water compared with those expressing high satisfaction (OR = 75.53, 95% CI: 23.11–246.90, *p* < 0.001). This strong correlation illustrates that dissatisfaction with municipal water quality directly drives a portion of the population to seek untreated alternatives, which increases their risk of waterborne diseases.

This correlation underscores the crucial role that public perception plays in water source choices and emphasizes the need for both improvements in water treatment infrastructure and efforts to increase public trust in the safety of municipal water. These findings, as illustrated in Figure 3 and Table 2, are particularly relevant, given the strong association between untreated water and increased diarrheal disease observed in this study.

The incidence and temporal distribution of diarrhea cases in the municipality of Southwestern Goiás were analyzed monthly and biannually from 2013 to 2019. Statistical analyses were performed to evaluate the seasonal trends and the relationship between the progression of cases over the years (Figure 4).

Diarrhea cases varied across months, with no statistically significant difference found between the months (Kruskal–Wallis *p* = 0.9338). The highest number of diarrhea cases occurred in March (mean = 1547), while the lowest incidence was observed in February (mean = 1221). Standard deviations indicated substantial variability in case numbers for each month, reflecting irregular patterns of incidence. Although March showed a higher mean number of cases compared with other months, Dunn’s multiple comparisons test did not reveal significant differences between any two months (all *p* values > 0.9999). This suggests that the monthly distribution of diarrhea cases may not follow a distinct seasonal pattern in this population (Figure 4A).

The comparison between the first and second semesters of each year did not show a significant difference in the number of diarrhea cases (Mann–Whitney U test *p* = 0.8257). The mean number of cases in the first semester was 1279, while in the second semester, it was 1325, with no significant variation between the two periods. The variability was higher in the second semester (coefficient of variation = 72.64%) compared with the first semester (coefficient of variation = 55.02%), indicating that case numbers fluctuated more in the latter half of the year. The analysis suggests that diarrhea incidence does not significantly differ between semesters, implying that other factors may be influencing the temporal distribution of cases (Figure 4B).

The temporal analysis using Pearson’s correlation showed a significant decline in the number of diarrhea cases over the study period (r^2^ = 0.8756, *p* = 0.00019). The slope of the regression line (−5519) indicates that there was a consistent decrease in the number of cases each year, with an estimated reduction of approximately 5519 cases per year. The analysis projected that, by 2021, the number of diarrhea cases would approach zero, as indicated by the x-intercept of 2021 (95% CI: 2019–2024). This finding reflects the successful interventions or improvements in public health practices during the study period, resulting in a marked decline in diarrhea incidence over time (Figure 4C).

The use of water from springs in 2016 and 2020 was evaluated, revealing significant changes in consumption patterns between these two periods. In 2016, 39% of the 383 households surveyed, amounting to approximately 149 households, reported using water from springs, while 61%, or around 234 households, did not. By 2020, this distribution shifted significantly, with only 2% of households (approximately 8 households) continuing to consume spring water, while 98% (about 375 households) reported no longer using this source. This marked decrease in spring water usage (*p* < 0.05) aligns with the closure of a key spring water collection site in 2017 (Figure 5).

The 2017 closure was a pivotal event, significantly reducing community access to untreated water and prompting a transition to municipally treated water. This shift in water consumption patterns is associated with a notable reduction in diarrhea cases, as untreated water sources like springs had a strong association with increased risks of waterborne diseases. The 96% reduction in the proportion of households using spring water from 2016 to 2020 underscores the impact of infrastructural changes on public water consumption habits and demonstrates the public health benefits of transitioning to safer, treated water sources.

The influence of the water source on diarrhea incidence was analyzed, examining the relationship between the use of treated versus spring water (Table 3). The data indicate a significant association between water source and diarrhea cases, with a higher incidence of diarrhea observed among households using spring water. During the period from 2013 to 2016, 149 cases were reported among those using spring water, compared with only 8 cases in the period from 2017 to 2019, following the closure of a major spring water collection site. This reduction in cases (*p* < 0.001) underscores the potential impact of public health interventions and infrastructure improvements on lowering diarrhea incidence.

Furthermore, the odds ratio analysis shows that households using spring water had an approximately ninefold increase in the odds of diarrhea occurrence (OR = 8.78; 95% CI: 4.37–18.29). While these results highlight untreated water as a potential risk factor for diarrhea, it is important to note that this ecological study does not establish a direct causal link between spring water usage and individual risk of diarrhea. Instead, the observed reduction in diarrhea cases after transitioning to treated water emphasizes the importance of reliable access to safe drinking water in mitigating waterborne disease incidence.

## 4. Discussion

The findings of this ecological observational study highlight the essential role of treated water in potentially reducing waterborne diseases, particularly diarrheal illnesses, in vulnerable populations. In Mineiros, Goiás, a marked reduction in diarrhea cases was observed from 2013 to 2019, with cases decreasing from approximately 3000 in 2013 to 1000 in 2019. This trend aligns with the population’s shift from untreated to treated water sources following the closure of the Magnífica Spring in 2017. This observation aligns with other studies that emphasize sanitation improvements as a critical factor in reducing disease incidence [6,16]. While this ecological approach does not establish causation, it suggests a plausible correlation between transitions in water sources and improved health outcomes, underscoring the importance of access to treated water to prevent infectious diseases.

The closure of the Magnífica Spring—a major untreated source widely used by the community—likely reduced exposure to waterborne pathogens at the population level. Odds ratio analysis (OR = 8.78) indicates that populations using untreated spring water had a nearly ninefold increase in the odds of diarrhea occurrence compared with those using treated water. While this finding supports the view that untreated sources can act as common vectors for pathogen transmission, particularly in low- and middle-income settings [1,4,5], it is important to interpret these odds ratios cautiously. Given the ecological design of this study, these associations reflect population-level trends rather than individual risk. Nonetheless, the results highlight how public health interventions that expand treated water access may correlate with a lower incidence of diarrhea on a community scale.

The municipal water samples met Brazilian safety standards, underscoring the efficacy of the treatment processes. In contrast, untreated sources like springs and streams frequently exceeded safety limits, particularly for fecal coliforms, emphasizing the need for a robust treatment infrastructure to minimize exposure risks. This observation aligns with broader research highlighting the importance of adherence to physicochemical and microbiological standards to prevent waterborne diseases [11,17]. Future studies may also examine the recreational use of untreated sources, as microbial contamination can impact human health, environmental quality, and biodiversity [20].

Turbidity, although commonly used as a water quality parameter, does not directly indicate microbial contamination. Instead, it reflects the presence of suspended particulates, which can influence bacterial adhesion and provide protection from disinfection processes [18]. Some studies have demonstrated that the composition of turbidity-causing materials plays a crucial role in microbial attachment and disinfection effectiveness [21,22]. However, turbidity alone is insufficient for determining microbial safety, as pathogens such as viruses and free-floating bacteria can persist in water with low turbidity [19]. Given these limitations, the present study employed fecal coliform quantification as a direct microbiological indicator, ensuring a more accurate assessment of microbial contamination. The correlation observed between turbidity and fecal coliform levels reinforces the need for an integrated approach that combines physical and microbiological indicators to improve water quality monitoring and risk assessment.

Only 10% of respondents expressed high satisfaction with the municipal water supply, suggesting a discrepancy between perceived and actual water quality. Sensory factors such as taste and odor may contribute to low satisfaction levels, which could impact adherence to treated water use even when quality standards are met [21]. Addressing these perceptual barriers is essential, as acceptance of treated water is critical to reducing reliance on potentially contaminated sources [1].

Another notable finding was the elevated aluminum levels in some treated water samples, occasionally exceeding the recommended limits. Prolonged exposure to aluminum has been associated with neurotoxic risks, including neurodegenerative diseases such as Alzheimer’s [22,23]. Ongoing monitoring and improvements in treatment processes are necessary to manage such contaminants. Additionally, certain samples had pH values outside the recommended range, underscoring the need for optimization of treatment processes to consistently meet water quality standards [24].

While this study provides important insights, several limitations inherent to its ecological design must be acknowledged. As an ecological study, it cannot establish direct causality between water quality and individual health outcomes. Furthermore, external factors that were not directly analyzed, such as sanitation practices and food safety, may also influence diarrhea incidence [25]. These factors are crucial for a comprehensive understanding of waterborne diseases and should be incorporated into future studies. Using mixed-method approaches that integrate both individual- and population-level data could reduce ecological bias and strengthen future research findings.

Future research may also examine a broader range of water contaminants, including pesticides and emerging pollutants, to provide a more holistic assessment of water-related health risks. These contaminants are increasingly relevant in water quality assessments, particularly in low- and middle-income regions, where comprehensive monitoring may be limited [26,27] Addressing these pollutants through targeted studies could guide public health interventions in similar settings.

In summary, this study adds valuable knowledge to our understanding of water quality impacts on health, emphasizing the need for sustained public health education, infrastructure improvements, and systematic monitoring. Collaborative efforts among public health officials, water management authorities, and communities are essential to sustainably address these challenges and ensure long-term public health. Investment in treated water infrastructure, coupled with community education on water quality, is fundamental to preventing waterborne diseases and protecting vulnerable populations [2,28]. The reduction in diarrhea cases in Mineiros following the closure of untreated sources underscores the importance of policies that expand access to safe, treated water in rural and peri-urban areas.

## 5. Conclusions

The findings from this study in Mineiros, Goiás, underscore the positive public health impact of transitioning from untreated to treated water sources. The observed reduction in diarrhea incidence from 2013 to 2019, particularly after the closure of the untreated Magnífica Spring in 2017, highlights the significant role of treated water in reducing waterborne diseases and safeguarding community health.

While improvements have been made in water treatment and public satisfaction, our findings point to the need for continued vigilance. Instances of elevated aluminum and fecal coliform levels, along with pH fluctuations, indicate areas for further enhancement of the treatment processes to ensure compliance with physicochemical and microbiological standards. These observations emphasize the necessity for sustained investment in water infrastructure and systematic quality monitoring to ensure consistent safety in drinking water.

This study also highlights the importance of collaboration among local authorities, health officials, and the community to address these challenges. Continued focus on water quality improvements and compliance with safety standards will be essential for maintaining public health gains, preventing disease outbreaks, and supporting the well-being of vulnerable populations over the long term.

## Figures and Tables

**Figure 1 ijerph-22-00436-f001:**
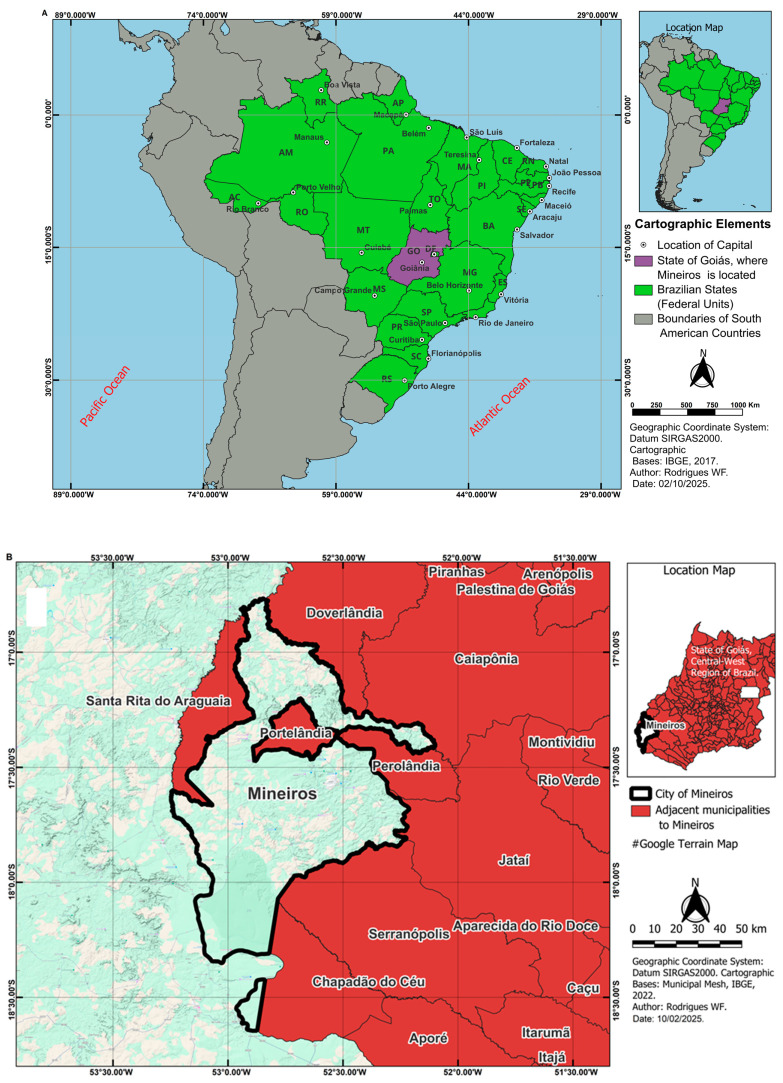
Geospatial representation of the study area. Panel (**A**) illustrates the geographical positioning of the state of Goiás within Brazil, South America, and its proximity to the Atlantic and Pacific Oceans. The scale bar ranges from 0 to 1000 km, with intervals of 250 km. Goiás is highlighted in purple, while the other Brazilian states are depicted in green. Neighboring countries that share a border with Brazil are represented in gray. The grid was inserted using degrees, minutes, and suffix notation. Panel (**B**) presents a detailed delineation of the target municipality, Mineiros, within the state of Goiás. The municipality is circled in black, while other cities included in the study are filled in red. The scale bar ranges from 0 to 50 km, with intervals of 10 km. Official cartographic coordinates were obtained from the Brazilian Institute of Geography and Statistics (IBGE) and referenced to the SIRGAS 2000 data. The dataset for Panel (**A**) corresponds to 2017, while for Panel (**B**), data from 2022 were used. The map was generated using QGIS version 3.34.15 (https://qgis.org/download/, accessed on 10 February 2025).

**Figure 2 ijerph-22-00436-f002:**
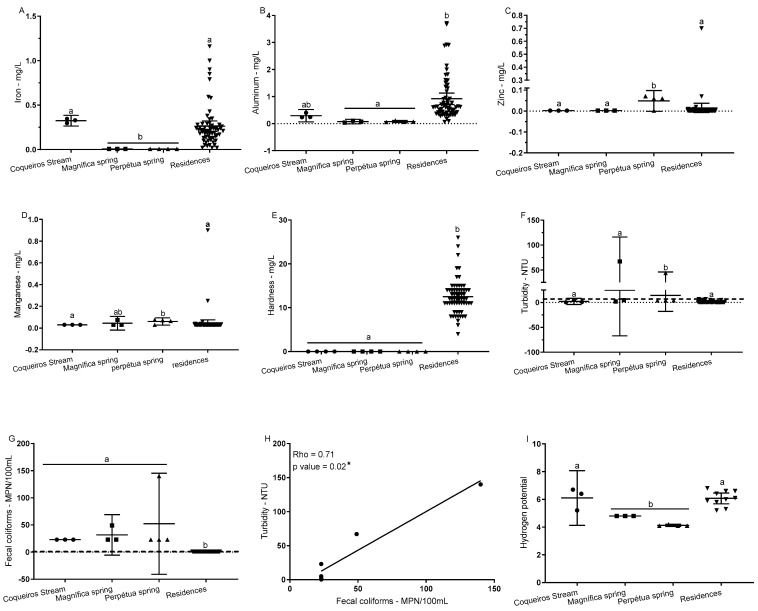
Effect of sampling sites on the concentrations of water quality parameters. Water samples were collected from four locations: Coqueiros Stream, Magnífica Spring, Perpétua Spring, and household taps. The evaluated parameters included iron (**A**), aluminum (**B**), zinc (**C**), manganese (**D**), hardness (**E**), turbidity (**F**), fecal coliforms (**G**), the correlation between turbidity and fecal coliforms (**H**), and pH (**I**). The Kruskal–Wallis test was used to compare parameters across sampling sites, followed by Dunn’s multiple comparison post-test to identify significant differences between specific locations. Different letters (e.g., “a” and “b”) denote statistically significant differences between sampling sites (*p* < 0.05). Spearman’s rank correlation test was used to analyze the relationship between turbidity and fecal coliform concentrations (**H**). The significance level for all analyses was set at 5%. * indicates *p*-value < 0.05.

**Figure 3 ijerph-22-00436-f003:**
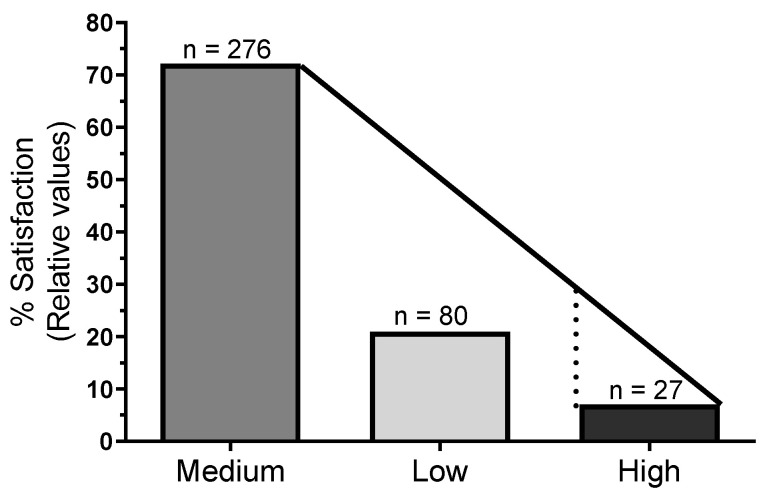
Distribution of satisfaction levels (low, medium, or high) with treated water provided by the municipality and its relationship with the use of untreated water sources (natural springs). “Satisfaction” reflects the population’s perception of the municipal water quality. The figure shows the number of individuals reporting different levels of satisfaction with treated water and their corresponding likelihood of using spring water.

**Figure 4 ijerph-22-00436-f004:**
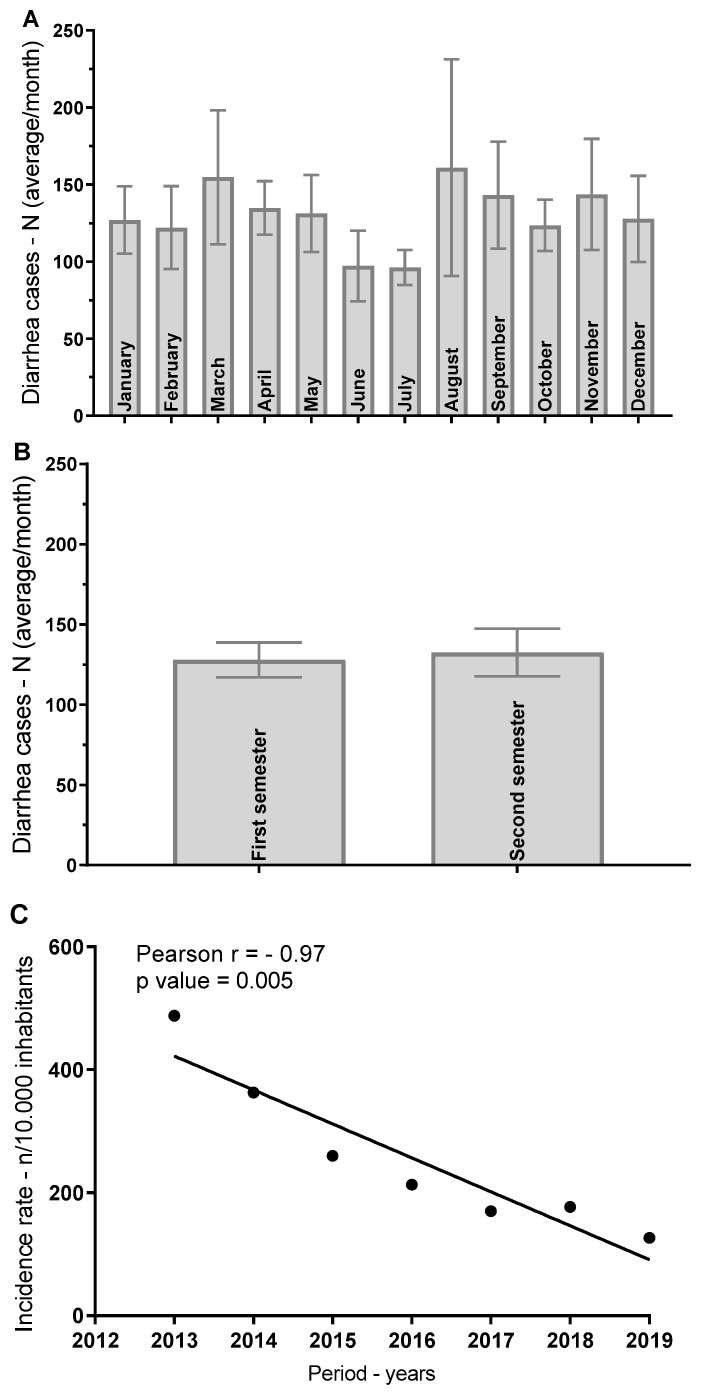
Incidence and temporal distribution of diarrhea cases in a southwestern Goiás municipality, Brazil (2013–2019): monthly and biannual analysis with Pearson’s correlation. (**A**) Monthly distribution of diarrhea cases. (**B**) Comparison of the number of diarrhea cases between the first and second semesters of each year. (**C**) Pearson’s correlation analysis of the yearly progression of diarrhea cases. The significance level used was 5%. Data were collected from monthly distributions between 2013 and 2019.

**Figure 5 ijerph-22-00436-f005:**
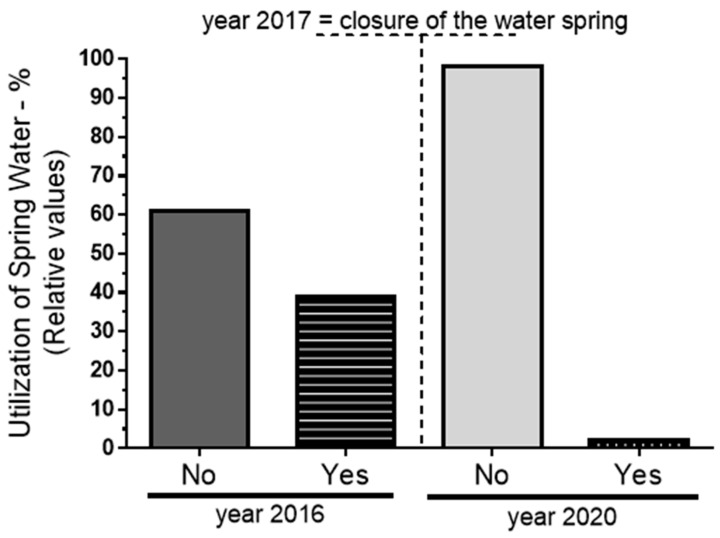
Change in water consumption from springs between 2016 and 2020. The data, obtained through a questionnaire, show a significant reduction in the percentage of respondents using spring water from 2016 (39%) to 2020 (2%), reflecting a major shift in consumption behavior following the closure of a key spring water collection site in 2017. The significance level used for the analysis was *p* < 0.05.

**Table 1 ijerph-22-00436-t001:** Comparison of the population’s satisfaction with water quality and usage of untreated spring water before and after the 2017 interdiction.

Until year 2016			
Satisfaction	Used spring water		Total—N (%)
	No—N (%)	Yes—N (%)	
High	49 (79.03)	13 (20.97)	62 (100)
Low	2 (1.15)	172 (98.85)	174 (100)
Medium	69 (46.94)	78 (53.06)	147 (100)
Total	120 (31.33)	263 (68.67)	383 (100)
χ^2^			155.90
*p* value			<0.001 *
Year 2017: interdiction of spring water			
Until year 2016			
Satisfaction	Used spring water		Total—N (%)
	No—N (%)	Yes—N (%)	
High	62 (100.00)	0 (0.00)	62 (100)
Low	28 (16.09)	146 (83.91)	174 (100)
Medium	100 (68.03)	47 (31.97)	147 (100)
Total	190 (49.61)	193 (50.39)	383 (100)
χ^2^			161.1
*p* value			<0.001 *

χ^2^ = chi-square test; * = statistically significant association (*p* < 0.005); N = absolute number; % = relative percentage (row value).

**Table 2 ijerph-22-00436-t002:** Logistic unadjusted regression analysis of the influence of satisfaction levels with treated water on the likelihood of using untreated spring water.

Until year 2016				
Predictor	Odds ratio	Lower	Upper	*p* value
Satisfaction				
Low–high	324.15	70.74	1485.30	<0.001 *
Low–medium	75.53	23.11	246.90	<0.001 *
Medium–high	4.26	2.13	8.51	<0.001 *
Year 2017: interdiction of spring water				
Until year 2020				
Predictor	Odds ratio	Lower	Upper	*p* value
Satisfaction				
High #	---------	---------	---------	---------
Low–medium	11.09	6.51	18.90	<0.001 *

Estimates represent the log odds of responses to the question “Do you use spring water? = Yes” versus “Do you use spring water? = No,” with confidence intervals (CI) and *p* values provided for each satisfaction level. “#” indicates the absence of individuals who reported using spring water. “*” indicates *p*-value < 0.05.

**Table 3 ijerph-22-00436-t003:** Comparison of diarrhea incidence before and after the transition to treated water in a municipality in southwestern Goiás, Brazil.

Total	Use of Spring Water	Ẋ Case/10,000 Hab.	Period
164	8	156	2017–2019
480	149	331	2013–2016
644	157	487	Total
49.3			X^2^
<0.001 *			*p* value
8.78			Odds ratio
4.37 to 18.29			95% CI

Ẋ = mean; Hab. = habitants; CI = confidence interval. * indicates *p*-value < 0.05.

## Data Availability

The datasets generated and analyzed during the current study, including the physicochemical aspects of water, water potability associations, and raw data for absolute frequencies of diarrhea cases, are openly available in the Open Science Framework (OSF) repository at https://doi.org/10.17605/OSF.IO/KYJ3A.

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
