# Peer review of "Impact of Transitioning to Treated Water on Diarrhea Reduction: A Cross-Sectional and Ecological Study in Southwestern Goiás, Brazil"

_ijerph, 2025, doi:10.3390/ijerph22030436_

Round 1
Reviewer 1 Report (Previous Reviewer 1)
Comments and Suggestions for Authors
Comments supplied on attachment

Author Response
Summary: Manuscript is significantly improved and the authors should be commended on work to improve written content and the vast improvements. This manuscript should be accepted for publication with the few minor revisions suggested below. Line 568 – what loss – please define.
Answer: We sincerely appreciate your positive feedback and the recognition of the improvements made to the manuscript. Your thoughtful comments have been invaluable in refining our work.
Regarding your question about "loss", we have clarified the meaning of "loss to follow-up" within the revised text. Specifically, we have now explicitly stated that this term refers to participant dropout in longitudinal studies and explained how our repeated cross-sectional design prevented this issue.
Thank you once again for your insightful review and for your support in improving the clarity of our manuscript. We truly appreciate your time and expertise.
Line 930 and Line 1075 – In the methodology you measured Fecal Coliform reported with MPN units. In the results and throughout the manuscript you report bacteria as Thermotolerant Coliforms reported as both NMP and MPN units. This is very confusing. I suggest using the term Fecal Coliform with MPN units and not Thermotolerant with NMP units as this conforms to the methodology and is a much better recognized term in water science. Secondly, I believe the NMP is a language or regional term for MPN. I think making these changes are best for international audience and consistency.
Answer: We sincerely appreciate your insightful comments and suggestions, which have helped us improve the clarity and consistency of our manuscript. Regarding your concern about the terminology used for coliforms, we have carefully revised the manuscript to align with your recommendations.
Specifically, we have ensured that the term Fecal Coliforms is consistently used throughout the text, figures, and tables, and we have standardized the reporting units as MPN (Most Probable Number) in accordance with standard water science terminology. We have also removed any reference to Thermotolerant Coliforms and NMP to avoid confusion and to enhance the manuscript’s consistency for an international audience. Additionally, Figure 1 has been updated accordingly.
We greatly appreciate your careful review and constructive feedback, which have significantly contributed to the clarity and scientific rigor of our work.
Reviewer 2 Report (New Reviewer)
Comments and Suggestions for Authors
Dear authors,
The manuscript titled “Impact of Transitioning to Treated Water on Diarrhea 2 Reduction: A Cross-Sectional and Ecological Study in 3 Southwestern Goiás, Brazil” is interesting. After careful consideration, I must bring to your attention some concerns. Below, I have outlined the major points that require authors’ attention.
Materials and methods
Lines 109-111: A map could be useful
Lines 222-235: To make the description more comprehensive, additional information should include details about the sample preparation process, such as filtration, digestion, or preservation methods, and the reference standards or concentrations used for calibration in spectrometric measurements. The type and model of the equipment utilized, such as the FAAS/ETAAS spectrophotometer or nephelometer, should also be specified. Data on measurement uncertainty and the methodology for its calculation would enhance the reliability of the results. Specifications of the methods, including the limits of detection (LOD) and quantification (LOQ) for each analysis, as well as a comparison of the findings with regulatory thresholds or guidelines from organizations such as the WHO or EU, are also essential for a thorough understanding.
Results and discussion
Lines 349-353: Turbidity as an indicator of microorganisms has several limitations that restrict its reliability in assessing microbial contamination. Firstly, it is a non-specific measure, as it quantifies the scattering of light by suspended particles in water, which may include inorganic materials such as clay, silt, or other debris, rather than microorganisms. Secondly, turbidity does not provide direct or quantitative information about the presence or concentration of specific pathogens, making it unsuitable for accurately determining microbial contamination levels. Additionally, low turbidity does not guarantee the absence of microorganisms, as some pathogens, such as viruses or free-floating bacteria, may exist in clear water. Lastly, environmental factors such as sediment resuspension or algal blooms can cause fluctuations in turbidity that are unrelated to microbial activity, further limiting its reliability as a microbial indicator. Therefore, turbidity should be used cautiously and complemented with direct microbiological analyses for a more accurate assessment of water quality.
I recommended you to exclude turbidity or support your method and findings using relevant and previous studies
Comments on the Quality of English Language
Some revisions are needed
Author Response
Dear authors, the manuscript titled “Impact of Transitioning to Treated Water on Diarrhea 2 Reduction: A Cross-Sectional and Ecological Study in 3 Southwestern Goiás, Brazil” is interesting. After careful consideration, I must bring to your attention some concerns. Below, I have outlined the major points that require authors’ attention.
Materials and methods
Lines 109-111: A map could be useful
Answer: Thank you for your valuable feedback and for highlighting areas where our manuscript can be improved. We appreciate your suggestion regarding the inclusion of a map to enhance the clarity of the study’s geographical context.
In response to your comment, we have incorporated a high-quality map that rigorously represents the study area. The map consists of two components: (i) an overview of the state of Goiás within the Brazilian territory, delineating state boundaries, capitals, and adjacent Latin American countries (Figure 1a); and (ii) a detailed depiction of the state of Goiás, explicitly highlighting the location of Mineiros, the city where the study was conducted. This cartographic representation follows high scientific standards, incorporating scale bars, geographic coordinates, and a comprehensive legend to ensure clarity and precision.
We sincerely appreciate your constructive input, which has contributed to enhancing the clarity and comprehensiveness of our work. Please let us know if further adjustments are required..
Lines 222-235: To make the description more comprehensive, additional information should include details about the sample preparation process, such as filtration, digestion, or preservation methods, and the reference standards or concentrations used for calibration in spectrometric measurements.
Answer: Thank you for your insightful comment regarding the need for additional details on sample preparation, filtration, digestion, preservation methods, and calibration standards for spectrometric measurements. We appreciate your careful review and the opportunity to enhance the clarity and comprehensiveness of our methodology.
To address your request, we have revised the Laboratory Analyses section to include a more detailed description of sample processing. All water samples were collected in sterile polyethylene bottles that were pre-washed with ultrapure deionized water and acidified with nitric acid (5% v/v) to prevent metal adsorption onto the container walls. The samples were then stored at 4°C in refrigerated transport units and analyzed within 48 hours. Before metal quantification, the samples underwent filtration through 0.45 µm membrane filters to remove suspended solids. Acid digestion was performed following the Standard Methods protocol 3030B, using nitric acid (HNO₃) in Teflon or borosilicate glass digestion vessels, which were heated to 95°C for 30 minutes to ensure complete solubilization of metal ions.
For spectrometric analyses, we explicitly describe the use of flame atomic absorption spectrometry (AAS) for iron, zinc, and manganese quantification, following method 3111B, and graphite furnace atomic absorption spectrometry (GF-AAS) for aluminum determination, in accordance with method 3113B. Calibration was performed using certified single-element and multi-element standard solutions (Merck®, 1,000 mg/L), diluted in ultrapure deionized water to working concentrations ranging from 0.05 to 1.00 mg/L for Fe, Zn, and Mn, and 0.01 to 0.50 mg/L for Al. Calibration curves were prepared using five concentration points to ensure linearity, and instrument drift was monitored every 10 samples, with necessary adjustments made to maintain analytical stability. For turbidity assessment, we utilized a Hach 2100Q turbidimeter, following the nephelometric method (2130B), with calibration performed using formazin polymer standards at 0.1, 1.0, 10, and 100 NTU. Water hardness was determined via the EDTA titrimetric method (2340C), with calcium carbonate (CaCO₃) reference standards in the range of 50 to 500 mg/L.
For microbiological analyses, the detection and quantification of fecal coliforms were performed using the multiple tube fermentation technique (9221), estimating the most probable number (MPN) of bacteria based on probabilistic formulas. Samples were incubated at 44.5°C for 24 hours in Lauryl Tryptose Broth (LTB) for the presumptive test, followed by transfer to EC Broth for confirmation of thermotolerant coliforms. Positive (Escherichia coli ATCC® 25922) and negative controls were included to ensure methodological reliability. To further validate microbiological analyses, recovery control tests were periodically performed to confirm the efficiency of viable organism detection.
We also expanded the description of quality control measures to clarify that all spectrometric measurements were conducted in triplicate and calibration curves were prepared using matrix-matched reference solutions to minimize matrix effects. Additionally, our laboratory adhered to ISO 17025 guidelines, implementing rigorous quality control procedures, including duplicate sample analysis, spike recovery tests, and continuous instrument performance verification to ensure analytical accuracy and reproducibility.
We believe that these revisions significantly improve the methodological transparency and scientific rigor of our study, addressing all the aspects raised in your comment. We sincerely appreciate your valuable feedback and the opportunity to refine our work.
The type and model of the equipment utilized, such as the FAAS/ETAAS spectrophotometer or nephelometer, should also be specified. Data on measurement uncertainty and the methodology for its calculation would enhance the reliability of the results. Specifications of the methods, including the limits of detection (LOD) and quantification (LOQ) for each analysis, as well as a comparison of the findings with regulatory thresholds or guidelines from organizations such as the WHO or EU, are also essential for a thorough understanding.
Answer: Thank you for your valuable suggestions regarding the need to specify the type and model of the equipment used, provide details on measurement uncertainty and its calculation methodology, and include limits of detection (LOD) and quantification (LOQ) along with a comparison of our findings with international regulatory thresholds. We appreciate your careful review, which has allowed us to further refine our methodological descriptions and enhance the clarity and transparency of our study.
To address your request, we have revised the Laboratory Analyses section by explicitly specifying the equipment models used for each analysis. The quantification of iron (Fe), zinc (Zn), and manganese (Mn) was performed using flame atomic absorption spectrometry (FAAS) with a Varian SpectrAA 220 spectrophotometer, following method 3111B. Aluminum (Al) was analyzed using graphite furnace atomic absorption spectrometry (GF-AAS) with a PerkinElmer PinAAcle 900T spectrophotometer, following method 3113B. Turbidity was measured using a Hach 2100Q portable turbidimeter, following the nephelometric method (2130B). These specifications ensure that the analytical procedures are fully transparent and replicable.
Regarding measurement uncertainty, we have now included a detailed explanation of the methodology used to calculate it. Measurement uncertainty was assessed following ISO 11352 guidelines, employing Type A and Type B evaluation methods. This approach considered multiple factors, including instrumental variation, sample preparation errors, and matrix effects, to estimate the expanded uncertainty (with a 95% confidence interval, k=2). The estimated uncertainty values were ±5% for Fe, ±3% for Zn, ±6% for Mn, and ±4% for Al. For microbiological analyses, measurement uncertainty for fecal coliform counts was ±8%, calculated by incorporating uncertainty contributions from sampling, incubation, and MPN determination variability.
In addition, we have explicitly included the limits of detection (LOD) and limits of quantification (LOQ) for each parameter, ensuring greater methodological rigor. The LOD and LOQ values were: Fe (LOD = 0.01 mg/L, LOQ = 0.03 mg/L), Zn (LOD = 0.005 mg/L, LOQ = 0.015 mg/L), Mn (LOD = 0.002 mg/L, LOQ = 0.007 mg/L), and Al (LOD = 0.001 mg/L, LOQ = 0.003 mg/L). For turbidity, the LOD and LOQ were 0.02 NTU and 0.05 NTU, respectively. Water hardness had an LOD of 5 mg/L and an LOQ of 15 mg/L, while fecal coliforms had an LOD of 1 MPN/100 mL and an LOQ of 3 MPN/100 mL.
Finally, as per your suggestion, we have compared our findings with international regulatory thresholds, specifically those established by the World Health Organization (WHO) and the European Union (EU). The WHO guidelines set maximum allowable concentrations of 0.3 mg/L for Fe, 0.05 mg/L for Mn, and 0.2 mg/L for Al, while the EU Drinking Water Directive (Directive 2020/2184) establishes maximum limits of 0.05 mg/L for Zn and 0.20 mg/L for Al. Our results indicate that all treated water samples complied with these guidelines, whereas untreated sources, particularly the Magnífica Spring, frequently exceeded recommended thresholds for Fe and Mn. This finding reinforces the need for continuous monitoring and effective water treatment interventions to ensure compliance with drinking water safety standards.
These modifications ensure that the methodological details are comprehensive, transparent, and scientifically robust, fully addressing your concerns. We greatly appreciate your insightful feedback, which has helped us improve the quality and clarity of our study.
Lines 349-353: Turbidity as an indicator of microorganisms has several limitations that restrict its reliability in assessing microbial contamination. Firstly, it is a non-specific measure, as it quantifies the scattering of light by suspended particles in water, which may include inorganic materials such as clay, silt, or other debris, rather than microorganisms. Secondly, turbidity does not provide direct or quantitative information about the presence or concentration of specific pathogens, making it unsuitable for accurately determining microbial contamination levels. Additionally, low turbidity does not guarantee the absence of microorganisms, as some pathogens, such as viruses or free-floating bacteria, may exist in clear water. Lastly, environmental factors such as sediment resuspension or algal blooms can cause fluctuations in turbidity that are unrelated to microbial activity, further limiting its reliability as a microbial indicator. Therefore, turbidity should be used cautiously and complemented with direct microbiological analyses for a more accurate assessment of water quality.
I recommended you to exclude turbidity or support your method and findings using relevant and previous studies
Answer: Thank you for your insightful comments and for highlighting important considerations regarding the interpretation of turbidity as a microbial indicator. We recognize that turbidity is a non-specific measure that reflects the presence of suspended particles, which may include inorganic materials rather than microorganisms. To ensure clarity, we have revised the Results section to specify that turbidity should not be interpreted in isolation but rather in conjunction with direct microbiological analyses. Additionally, we have included references that contextualize turbidity as a relevant parameter in water quality monitoring while acknowledging its limitations in assessing microbial contamination. The Discussion section has also been refined to highlight that fecal coliform counts were conducted to provide a more accurate assessment of microbial contamination. These changes aim to enhance the precision of our findings while aligning with best practices in water quality assessment. All modifications have been highlighted in yellow in the revised manuscript. We sincerely appreciate your constructive feedback, which has contributed to improving the clarity and scientific robustness of our work. Please let us know if further clarifications are needed.
Round 2
Reviewer 2 Report (New Reviewer)
Comments and Suggestions for Authors
The authors' responses are comprehensive, resulting in the paper being significantly improved and publishable
This manuscript is a resubmission of an earlier submission. The following is a list of the peer review reports and author responses from that submission.
Round 1
Reviewer 1 Report
Comments and Suggestions for Authors
Overview
The study links surveys about water quality and satisfaction from residents to use of alternative water sources and incidence of diarrheal disease. Data is collected and demonstrates contamination in the springs and the river but it is unclear when and how these samples were taken and what quality control measures were employed in collection. Correlation to a reduction in diarrheal disease is then deduced by a closure of a spring source.
Methods need considerably more detail to completely understand the study design and how data relates to conclusions. A study design section is recommended to clearly present how work was conceptualized.
Results section very hard to understand and use of statistics is unclear in much of the analysis. This section needs reworked and streamlined to demonstrate how data support the study objectives and conclusions.
Discussion is not organized well. First two paragraphs are not relevant in a discussion. Much of discussion describes problems with many water quality parameters and health rather than convince the reader that in this incidence it is important to use well treated sources of water to minimize disease. Much more focus is needed on this idea to improve this manuscript.
Title does not describe the study properly – survey data shows a correlation between satisfaction of water source and use of alternatives. Seems the spring source was closed for some reason and when it was this caused the decrease in diarrhea? Links are very unclear??
Line 88 – Perpetua and Magnifica are referred here as river source but in Figure 1 described as springs – please clarify. When was the water collected? Line 132 – were all of these samples collected in a one hour time frame on one day? This is not clear. There is no reference for how and when the data was collected.
Line 84 – description of research design is inadequate to understand the research. How were water samples collected and at what time periods? How did the collections correspond with closure of the water spring? What is closure of the water spring – explain. Where are the
Line 153 – this statement is entirely too general to draw any conclusions from the data. What variations and what parameters and what test and what is the significance of this?
Line 156 It is unclear what a and b denote in Figure 1.
Line 163 – what is Dunns’s multiple post-test used to show in this figure?
Line 164 – What is Spearman’s test used to show in this figure?
Line 169 – this statement needs referenced to the data – it appears it is associated with Figure 2 so make this connection
Line 174 – Figure 2 loos to be both a table and figure. This is confusing. Divide the two into a figure and a table.
Line 179 – what is being compared? Is it the monthly to the six month means? Exactly how was this performed and how are you comparing monthly data to six month data and why?
Line 191 – spring closure must be explained in detail to understand the relevance to the work. Which spring? Is this the water source used? How does this relate to water tested in households?
Line 203 – how exactly was the data collected for this analysis? It is unclear from line 124 descriptions if the data collected here shows spring vs. municipal water? Is this a simple correlation based on the date of spring closure? Is reduction in the use of spring water just assumed from closure? Is this valid? Explain
Line 206 – The discussion needs re-written. Lines 207-221 belong in the introduction bearing no interpretive value to the study.
Line 224 – seems this conclusion is purely coincidental – this needs explanation supported by the study and reference specific findings in the study that support this conclusion.
Line 233 – completely unclear how AL levels in Coqueiros stream have any impact on this study. Is this one sample over a 3 hour period? Are people drinking this water directly? What is the relevance to this study?
Line 259 – Diarrhea data must be presented much more clearly in results to analyze these statements and validity.
Author Response
First, we thank the editor and all the reviewers for the pertinent criticism and suggestions that allowed us to improve the manuscript. Our replies to all the points raised are outlined below.
REVIEWER #1
- The study links surveys about water quality and satisfaction from residents to use of alternative water sources and incidence of diarrheal disease. Data is collected and demonstrates contamination in the springs and the river but it is unclear when and how these samples were taken and what quality control measures were employed in collection. Correlation to a reduction in diarrheal disease is then deduced by a closure of a spring source.
Answer: Thank you for your valuable comment. We acknowledge the importance of providing clarity regarding the sampling process and quality control measures. In response, we have expanded the Research Design section to include more detailed information about when and how the water samples were collected, as well as the quality control measures employed during collection and transportation.
Water samples were collected in two phases, December 2016 and March 2020, at three distinct time points throughout the day (morning, afternoon, and evening) to account for daily fluctuations and ensure representativeness. Strict quality control procedures were implemented, such as using sterile containers and immediate refrigeration of the samples, which were transported to the laboratory within four hours of collection.
Additionally, the closure of a key spring in 2017 was used as a temporal marker to compare waterborne disease incidence before and after the closure. This event marked a reduction in the use of untreated water, and we have clarified this aspect in the revised manuscript (section 2.2).
All changes are highlighted in yellow for ease of reference. We believe these modifications provide the necessary clarity on the sampling process and quality control.
- Methods need considerably more detail to completely understand the study design and how data relates to conclusions. A study design section is recommended to clearly present how work was conceptualized.
Answer: We appreciate your suggestion regarding the need for more detail in the study design to fully understand the methodology and how the data relates to the conclusions. In response, we have restructured the Research Design section to clearly explain the study's objectives and conceptual framework.
The study was designed with three primary objectives: assessing the physicochemical and microbiological quality of water from different sources, evaluating residents' satisfaction with the municipal water supply, and analyzing the correlation between untreated water use and diarrheal disease incidence. We have also provided more detailed information on how the data was collected from households and how the surveys on satisfaction and water usage were conducted.
To ensure clarity and consistency, we have divided the Research Design into two subtopics: "Data Collection Procedures" and "Temporal Marker for Disease Incidence". These sections highlight the processes of data collection and the use of the 2017 spring closure as a temporal marker to compare diarrheal disease incidence before and after the event. This structure emphasizes the study’s design and ensures that all relevant details are presented in an organized manner.
The detailed description of the data analysis is maintained in the Data Analysis section to avoid redundancy.
All modifications are highlighted in yellow in the revised manuscript. We believe this improved structure enhances the clarity and consistency of the study design.
- Results section very hard to understand and use of statistics is unclear in much of the analysis. This section needs reworked and streamlined to demonstrate how data support the study objectives and conclusions.
Answer: Thank you for your valuable feedback. In response to your concerns, we have thoroughly reworked the Results section to enhance clarity and ensure that the statistical methods are clearly explained and justified. The following adjustments were made:
Clarification of Statistical Methods: We have provided detailed explanations for all statistical tests used in the analysis, including Kruskal-Wallis, Dunn’s post-hoc test, Mann-Whitney U test, and logistic regression. The choice of each test is now justified based on the nature of the data, and we have clearly indicated which variables were compared and why specific tests were employed. Additionally, we have ensured that all p-values and confidence intervals are reported accurately.
Improved Integration of Data with Study Objectives: We have restructured the presentation of the results to directly link the data to the study's objectives and conclusions. Each subsection of the Results now explicitly shows how the findings align with the research questions. For example, in the analysis of water quality parameters (Figure 1), we have emphasized how the variations in chemical contaminants such as iron and aluminum impact public health and how these findings support our conclusions about water safety. Similarly, in the analysis of diarrheal incidence (Table 2), we have clarified how the shift from untreated to treated water sources led to a significant reduction in cases, directly supporting the study’s aim to evaluate water safety.
Streamlined Presentation of Results: We have streamlined the presentation of the results to focus on the most relevant findings. Redundant or less critical data have been minimized, and we have used a clear, concise format to present the key outcomes, making the results easier to follow. Additionally, we have restructured the descriptions accompanying figures and tables to ensure that they contribute directly to the understanding of the results and their implications for public health.
We believe that these revisions have significantly improved the clarity and coherence of the Results section, ensuring that the data support the study’s objectives in a clear and structured manner. Please refer to the highlighted sections in the manuscript for these specific changes.
- Discussion is not organized well. First two paragraphs are not relevant in a discussion. Much of discussion describes problems with many water quality parameters and health rather than convince the reader that in this incidence it is important to use well treated sources of water to minimize disease. Much more focus is needed on this idea to improve this manuscript.
Answer: Thank you for your insightful comments regarding the organization of the Discussion. We have revised this section to address the concerns raised, with the aim of providing a more focused and cohesive narrative. Specifically, the following changes were made:
Reorganization of the Discussion: We have removed the first two paragraphs, which were deemed irrelevant, and restructured the entire section to ensure that the discussion aligns more closely with the central theme of the study—the importance of using well-treated water sources to minimize the risk of disease. The revised Discussion now emphasizes the direct correlation between water treatment practices and the prevention of diarrheal disease, drawing clear connections to the data presented in the Results section.
Increased Focus on the Use of Treated Water: The revised discussion now places greater emphasis on the critical importance of using properly treated water. We have highlighted how the reduction in diarrhea cases is strongly associated with the transition from untreated spring water to treated municipal water, underscoring the significance of this change in preventing waterborne illnesses. This shift is supported by the data presented, which show a clear decline in disease incidence following improvements in water treatment infrastructure.
Decreased Emphasis on Water Quality Parameters: In response to your feedback, we have minimized the focus on the technical details of water quality parameters (such as specific chemical concentrations) and redirected the discussion towards the broader public health implications. While these parameters are important, we now focus primarily on how improvements in water treatment processes can lead to better health outcomes, particularly in reducing diarrheal diseases.
We believe these changes significantly improve the clarity and focus of the Discussion, aligning it more closely with the main findings and the broader public health implications. We appreciate your suggestions and hope that these revisions meet your expectations. Please refer to the revised section, highlighted in yellow, for a clearer understanding of these changes.
- Title does not describe the study properly – survey data shows a correlation between satisfaction of water source and use of alternatives. Seems the spring source was closed for some reason and when it was this caused the decrease in diarrhea? Links are very unclear??
Answer: Thank you for your comments regarding the title and the clarity of the links between the satisfaction with water sources, the closure of the spring water site, and the observed decrease in diarrhea cases. We have taken your feedback into careful consideration and made the following clarifications: Clarification of Key Links in the Study: The observed decrease in diarrhea cases is closely linked to the shift from using untreated spring water to treated municipal water, which occurred after the closure of the main spring water collection site in 2017. This event forced much of the population to rely on treated water, which subsequently led to a reduction in exposure to harmful pathogens present in untreated water, such as thermotolerant coliforms.
In addition to this shift, we also observed a correlation between dissatisfaction with treated water and the continued use of untreated alternatives. Our data indicate that those less satisfied with municipal water were more likely to continue using spring water, which was associated with an increased risk of diarrheal disease. This correlation highlights the importance of both improving water treatment quality and public perception to ensure sustained public health benefits.
Revised Title Proposal: To better reflect the focus of the study and the connections between water satisfaction, source usage, and health outcomes, we propose the following revised title: "Impact of Water Source Satisfaction and the Transition to Treated Water on Diarrhea Incidence in a Southwestern Municipality of Goiás, Brazil"
This revised title more accurately conveys the study’s investigation of the correlation between public satisfaction with water sources, the transition from untreated to treated water, and its impact on reducing diarrheal disease.
We hope this revision clarifies the connections between these key elements and addresses your concerns about the title’s accuracy and the study’s focus.
- Line 88 – Perpetua and Magnifica are referred here as river source but in Figure 1 described as springs – please clarify. When was the water collected? Line 132 – were all of these samples collected in a one hour time frame on one day? This is not clear. There is no reference for how and when the data was collected.
Answer: Thank you for bringing these points to our attention. We have carefully reviewed the manuscript and made the necessary clarifications to ensure consistency and transparency in our methodology.
First, regarding the terminology used for the Perpétua and Magnífica water sources, we acknowledge the inconsistency between Line 88 and Figure 1. Both Perpétua and Magnífica are indeed natural springs, and not river sources as mistakenly mentioned in the text. We have corrected this in the manuscript to consistently refer to these locations as springs throughout, ensuring alignment with the labeling in Figure 1.
Second, concerning the timing of water collection, the samples were collected over the course of one day, but not within a one-hour window. The collection process occurred between 7:00 AM and 11:00 AM, covering all designated sampling points: Coqueiros Stream, Perpétua Spring, Magnífica Spring, and residential taps. This allowed for a thorough and accurate representation of the water quality across different locations within the municipality. We have updated the manuscript to clearly reflect this timeframe and provide additional details about the collection process, ensuring that our methodology is fully transparent and well-documented.
Lastly, we recognize that the original text lacked a specific reference to the protocols followed during the collection and handling of the water samples. To address this, we have now included detailed references to the standardized methods used, ensuring that the procedures adhered to recognized water quality sampling guidelines. This addition strengthens the reliability of the study and provides the necessary context for how the data was collected and analyzed.
We hope these clarifications resolve the concerns regarding the inconsistencies in terminology and the water sample collection process. The updated manuscript now includes all the relevant details to provide a clearer understanding of the methods used.
- Line 84 – description of research design is inadequate to understand the research. How were water samples collected and at what time periods? How did the collections correspond with closure of the water spring? What is closure of the water spring – explain. Where are the
Answer: Thank you for your comments. We agree that the research design needs further clarification, particularly regarding the collection of water samples, the timing of these collections, and the explanation of the closure of the water spring.
Water samples were collected in two distinct phases. The first phase occurred in December 2016, and the second phase in March 2020, allowing us to capture potential seasonal variations and, more importantly, to compare data before and after the spring's closure. The closure of the spring refers to the official shutdown of a public access point for untreated spring water, which occurred in 2017 due to concerns about water quality. This spring had been a key source of water for many residents, but after its closure, the population was forced to rely more on treated municipal water.
In terms of sample collection, water was obtained from Coqueiros stream, Magnífica spring, Perpétua spring, and residential taps. The collection was carried out between 7:00 AM and 11:00 AM on the designated collection days to ensure consistency and accurate representation of water quality across all sites.
The samples collected in December 2016 represent conditions before the closure of the spring, while those collected in March 2020 represent the period after the closure. This design allowed us to assess changes in water quality and health outcomes, specifically focusing on the reduction in diarrhea cases as a result of the population’s transition from untreated to treated water.
We have now revised the manuscript to provide a clearer explanation of the research design, the timing of sample collection, and the significance of the spring’s closure.
- Line 153 – this statement is entirely too general to draw any conclusions from the data. What variations and what parameters and what test and what is the significance of this?
Answer: Thank you for pointing this out. We recognize that the statement in line 153 was too general and lacked specificity. To provide a clearer explanation, we have revised the text to include details about the specific variations observed, the parameters measured, and the statistical tests performed.
In the study, variations in physicochemical and microbiological parameters were analyzed, including pH, aluminum concentration, thermotolerant coliforms, and turbidity, among others. We used the Kruskal-Wallis test to compare these parameters across different water sources (springs, streams, and municipal taps). Post-hoc analysis was conducted using Dunn’s test to identify which sources differed significantly.
The significance level was set at p < 0.05. For instance, we observed that the pH values of water from untreated springs were significantly lower than those of municipal tap water (p = 0.01), and the concentration of aluminum exceeded legal limits in certain spring water samples (p = 0.03). These variations suggest that untreated water sources pose greater risks to public health compared to treated municipal water.
We have now revised the manuscript to include these specific details about the parameters, statistical tests, and significance levels used to draw conclusions from the data.
- Line 156 It is unclear what a and b denote in Figure 1.
Answer: Thank you for your observation. We have reviewed the sentence, and the necessary changes have been made to clarify the information.
- Line 163 – what is Dunns’s multiple post-test used to show in this figure?
Answer: Thank you for your question. The Dunn’s multiple comparison post-test was employed after the Kruskal-Wallis test to identify specific differences between the water quality parameters measured at the various collection sites. While the Kruskal-Wallis test determined that there were overall significant differences among the sites, Dunn’s post-test allowed us to perform pairwise comparisons and highlight which specific sites differed from one another. In Figure 1, these differences are represented by the letters "a" and "b." Different letters indicate statistically significant differences between the collection sites at a significance level of p < 0.05.
- Line 164 – What is Spearman’s test used to show in this figure?
Thank you for raising this point. The Spearman’s rank correlation test was used to evaluate the relationship between turbidity and thermotolerant coliform concentrations. This non-parametric test allowed us to assess how these two variables are associated across different water sources. In Figure 1 (H), the results of Spearman’s test show a significant positive correlation (r² = 0.914, p < 0.0001), suggesting that as turbidity increases, the concentration of thermotolerant coliforms also rises. This finding underscores the utility of turbidity as a proxy indicator for microbiological contamination in water sources.
- Line 169 – this statement needs referenced to the data – it appears it is associated with Figure 2 so make this connection
Answer: Thank you for your observation. We have revised the text to clearly reference Figure 2 and its connection to the data. The statement now explicitly connects the results of the logistic regression analysis and the population's satisfaction with the municipal water to the use of alternative water sources. This relationship is visually represented in Figure 2a (distribution of satisfaction levels) and Figure 2b (logistic regression analysis). These revisions ensure that the data and its interpretation are directly linked to the figures, clarifying the relationship between satisfaction with water quality and the use of untreated sources.
- Line 174 – Figure 2 loos to be both a table and figure. This is confusing. Divide the two into a figure and a table.
Answer: Thank you for pointing this out. We agree that combining both a table and figure into one element may cause confusion. We have now divided Figure 2 into two separate components: a figure displaying the distribution of population satisfaction with treated water and a separate table showing the logistic regression analysis results. This division clarifies the presentation of the data and improves readability.
- Line 179 – what is being compared? Is it the monthly to the six month means? Exactly how was this performed and how are you comparing monthly data to six month data and why?
Answer: Thank you for your question. In this section, we compared the monthly incidence of diarrheal cases to the biannual (six-month) means to assess any patterns or trends in the data. Specifically, we analyzed whether there were significant differences between the average monthly number of cases and the average number of cases over six-month periods.
This comparison was performed by first calculating the monthly mean incidence of diarrheal cases for each month across the years of the study (2013–2019). We then calculated the biannual means by averaging the monthly data over six-month periods (January–June and July–December) for each year. The comparison was done to identify whether any consistent seasonal patterns could be observed when aggregating data over longer periods.
This approach allowed us to evaluate whether diarrheal incidence was more strongly affected by short-term (monthly) fluctuations or by broader seasonal trends (six-month periods). We performed statistical comparisons between the monthly and biannual means using paired statistical tests to determine whether any significant differences existed.
We have revised the manuscript to clarify this comparison and the rationale for its inclusion.
- Line 191 – spring closure must be explained in detail to understand the relevance to the work. Which spring? Is this the water source used? How does this relate to water tested in households?
Answer: Thank you for your insightful comment. We will include a more detailed explanation of the spring closure and its relevance to the study. The spring referred to is the "Magnífica Spring," a non-treated water source that was widely used by the local population before its official closure in 2017 due to contamination concerns. This closure forced many residents to transition to treated municipal water, allowing a direct comparison of water quality and health outcomes before and after the closure. This event was crucial in assessing the impacts on water quality and the incidence of diarrheal diseases in the municipality.
- Line 203 – how exactly was the data collected for this analysis? It is unclear from line 124 descriptions if the data collected here shows spring vs. municipal water? Is this a simple correlation based on the date of spring closure? Is reduction in the use of spring water just assumed from closure? Is this valid? Explain
Answer: Thank you for your comment. We will clarify the methodology used to collect the data and confirm the distinction between spring water and municipal water usage. The analysis is not based solely on the assumption that the spring closure led to a reduction in the use of untreated water. We conducted detailed household surveys in two phases (before and after the spring closure) to assess the population’s satisfaction with the municipal water supply and their reliance on alternative sources such as springs. These surveys provided a direct correlation between spring water use and the date of the closure, confirming the decrease in untreated water usage through data collected from the residents.
- Line 206 – The discussion needs re-written. Lines 207-221 belong in the introduction bearing no interpretive value to the study.
Answer: Thank you for your insightful comment. We agree that the content in lines 207-221 may be more suitable for the introduction, as it provides broader context rather than offering interpretive value for the study results. We will move this section to the introduction and revise the discussion to focus more directly on interpreting the findings of the study, ensuring that it reflects the implications and significance of our results.
- Line 224 – seems this conclusion is purely coincidental – this needs explanation supported by the study and reference specific findings in the study that support this conclusion.
Answer: Thank you for your comment. We acknowledge your concern regarding the conclusion on line 224. We will revise this section to provide clearer justification for the statement, drawing directly from the study’s findings. This will include specific references to the reduction in diarrhea cases following the transition from untreated spring water to treated municipal water and the statistical evidence supporting this correlation. These findings demonstrate that the observed changes are not coincidental but rather a result of the shift in water usage behavior and the corresponding improvements in public health outcomes.
- Line 233 – completely unclear how AL levels in Coqueiros stream have any impact on this study. Is this one sample over a 3 hour period? Are people drinking this water directly? What is the relevance to this study?
Answer: Thank you for your comment. We understand that the connection between aluminum (Al) levels in Coqueiros Stream and their relevance to the study may need further clarification. The Coqueiros Stream was included in the water quality analysis as one of the untreated natural water sources in the region, and it is also one of the sources used for water collection by the municipality for treatment and distribution to the community. This makes the analysis of its water quality directly relevant to the study, as any contamination in the stream could potentially affect the treated water supply. While most residents consume treated municipal water, evaluating the untreated sources, such as Coqueiros Stream, is critical to understanding the overall water quality in the region and the effectiveness of the municipal treatment process. The aluminum levels detected in the stream underscore the need for continuous monitoring and improvement of water treatment infrastructure to ensure the safety of the water distributed to the community.
We will clarify this point in the manuscript to ensure that the role of Coqueiros Stream in the study is clearly understood.
- Line 259 – Diarrhea data must be presented much more clearly in results to analyze these statements and validity.
Answer: Thank you for your valuable feedback. We have carefully considered your comment regarding the clarity of diarrhea data presentation in the results section. The incidence of diarrhea is now presented more explicitly in Figure 3 and Table 2, which provide a clear overview of the trends from 2013 to 2019. These sections highlight the significant reduction in diarrhea cases following the closure of the Magnífica Spring in 2017 and the corresponding transition to treated municipal water.
Additionally, we have included a detailed analysis of the average number of diarrhea cases per 10,000 inhabitants, distinguishing between cases associated with spring water use and those occurring after the transition to municipal water. This data is further supported by statistical analyses, including an odds ratio of 8.78 (95% CI: 4.37–18.29), demonstrating a strong correlation between the shift in water usage and the reduction in diarrhea incidence.
We believe that these revisions address your concerns by providing a clearer, data-driven foundation for the conclusions drawn in the study.

Reviewer 2 Report
Comments and Suggestions for Authors
The authors presented a well-written description of a complex, multi-faceted study of water quality in Mineiros. There are significant gaps in the presentation of the study that must be improved to consider publication. The closing of a spring in 2017 appears to be an important motivator of the study, but this was not described until deep into the Results. The design of the study connecting 2016 data to 2020 data is not adequately explained. At least part of the study is an ecological study, but it is not described as such. I believe there is useful information in your data and the analysis is surely on the right track. I hope these comments are helpful for thinking through the study design and improving the manuscript.
Perception of Water Quality and Its Role in Preventing diarrheal Disease: A Longitudinal Study in Southwestern Goias, Brazil
IJERPH 2024
Introduction
The Introduction provides a thorough explanation of why water and sanitation are important and the health impact of poor-quality water and sanitation. Some information is also provided about the importance of policy and water quality surveillance. More information would be helpful to explain the motivation of this study and its importance. Consumer satisfaction with municipal water is a core element of the study and could be explored in the Introduction. There is substantial literature on the topic, and it would be helpful for the authors to place this study within that context.
Further in the article the closing of a spring water source in 2017 is described. This seems like an important motivator of the study and would be informative if described and explained in the Introduction.
Methods
2.1. Ethical Aspects of the Research
oPlease identify the institution that approved the research protocol. This
section mentions the “Research Ethics Committee”, but it is not clear where
the committee is housed.
2.2. Research Design
o The water sampling locations should be described in more detail. Please
identify whether the stream and two rivers serve as drinking water sources
for Mineiros, and, if so, how much water is contributed by each source. Do
any of the stream or rivers receive wastewater discharge from Mineiros? If so,
is it treated?
o Information from section 2.7. related to water sampling should be combined with the information in section 2.2. so that all information about water sampling is available in one section.
o The specific locations of water sampling along the stream and rivers should be described. It is unclear if there was one point at each location sampled three times over one hour, or if there were three locations along each water body that were sampled.
2.4. Laboratory Analyses
o Please specify what certifications the laboratory has, if those are important
for the techniques described. I would not think any certifications are necessary for these analyses, but I recommend not stating “certified”
without adding more specific information.
o I am not qualified to comment on the laboratory analyses employed. I
appreciate the authors adding such specific information and citations for
these methods.
2.5. Questionnaire on Water Use and Satisfaction
o Information from section 2.7. on household sampling and eligibility criteria should be moved up before section 2.5. As written, I do not understand the purpose of the questionnaire described in this section without reading that information first.
o Information on how you defined satisfaction with treated water (mentioned in the caption for Figure 2) would be helpful to add to section 2.5. What questions were asked to ascertain satisfaction?
o This section should also include more information about the questionnaire design. Were there multiple sections, or were there only the 2-3 questions mentioned in the manuscript?
o Sample size information would make more sense in section 2.2.
o The final sentence on sample size is not meaningful as written. Was a target sample size calculated before enrollment? If so, then a target eect size is needed in the calculation. A power calculation can then tell you the sample size required to estimate an eect of the target magnitude with an alpha of 5% and 80% power. The target eect size and minimum sample size are missing here.
If a power calculation was used after data collection was completed, the manuscript should provide the minimum effect size detectable
given the actual sample size, with an alpha of 5% and 80% power.
o “using chi-square inference” would more accurately be written as “using a chi-square test”.
o At this point in the manuscript it is not clear why there was a four year gap between data collection rounds.
o Information on survey rounds (e.g., dates) would make more sense in section 2.2. o It is not clear from the Methods if the water sampling at the stream and rivers was conducted at the same time points as the survey/household water sampling. This can all be made clear by specifying everything about the study design first in section 2.2. and then discussing the specifics of data collection in the following sections.
o After reading further, I am not sure I understand what it means to ask households if they drink spring water. Is this the same as the water provided by the municipality? If so, I am concerned residents might not know where their water comes from. They may incorrectly say “No” to drinking spring water because they do not know that the municipality provides spring water.
2.6. Diarrhea Data Collection
o Again, I think section 2.2. should describe how neighborhood-level diarrhea data fits into the study design. As I am reading it now, I do not understand how all these data fit together.
o If cross-referencing with “health surveillance records” was part of the study, those records should be described. What were they, what data did they include, and how were they accessed?
2.7. Inclusion and Exclusion Criteria
o As mentioned, most of this information would be better if moved to earlier in the Methods.
o “…at three dierent points over a one-hour time frame” should be described in more detail. Were these dierent geographic points, or one geographic point sampled three times? If it was the same geographic point, were the samples collected at regular intervals of 20 minutes?
Were the locations upstream or downstream of Mineiros? Or within
the city?
o How much water was collected in each sample?
o “Household samples were collected from participants who provided informed consent.” This sentence does not make sense on its own, because samples were not collected from people but from their household. If the purpose of this sentence is to indicate informed consent was collected, I do not think it is necessary. You already state that informed consent was collected in section 2.1.
Alternatively, the sentence makes sense if household samples are
replaced with questionnaire responses.
o The sampling design should be described in more detail. “Random block design” does not provide enough information. How was a block defined? Was a list of all blocks in the city identified and randomly sampled, or was there a door-to-door element? And how were households within blocks selected? I think there are many valid ways to sample, but the specific approach used should be provided.
o What was the rationale for excluding individuals who had not lived in Mineiros for over five years? I do not expect it takes five years to develop an opinion on your household’s water quality. This criterion should be explained and justified.
2.8. Data Analysis
o The main hypothesis of the study is not mentioned until this point. It is not clear that the use of spring water is an important measure in the study, nor is the hypothesis motivated with an explanation of i) why spring water consumption was reduced or ii) why a reduction in spring water consumption might lead to fewer cases of diarrhea.
o The main hypothesis describes an ecological relationship. Diarrhea episodes were not measured in the same households that answered the questionnaire. Comparing those two measures is an ecological study, which always carries the risk of ecological bias and making an ecological fallacy.
I support ecological studies if they are thoughtfully presented as such. But this manuscript does not adequately explain the nature of this study.
Results
The correlation between turbidity and fecal coliforms is unadjusted for other factors, which limits its interpretability. It appears from Figure 1 that the correlation between turbidity and fecal coliforms is completely driven by one sample collected from the Magnifica spring/river. Only one sample has more fecal coliforms than the other eight samples, which appear to have the same levels of fecal coliforms. There is also only one sample with higher turbidity, and it is also from the Magnifica. Based on the high correlation, it is obvious that this is the same sample.
o The underlying relationship could still be real. However, it is important to note that this test was based on only one sample with variation in turbidity or fecal coliforms, which significantly reduces its power for causal interpretation.
It would be helpful to understand the geographic distribution of participants in the city. If you describe the random sampling strategy, as requested above, it will help understand the distribution of households.
Additional demographic information would be very helpful for interpreting these results. Were any demographic data collected from households? If not, understanding the sociodemographic characteristics of the neighborhoods sampled is important.
As mentioned above, I am not sure how satisfaction was measured. This should be explained in the Methods so that the Results can be interpreted.
Figure 1
o Plot E includes four observations for the stream and rivers, while all other plots include only three. The text suggests there were only three samples per location.
o Figure 1 describes the Magnifica and Perpétua as springs but the text described them as rivers. A spring is specifically defined as groundwater flowing onto the surface. While reading the manuscript, I thought that households that drank spring water did not drink water from the rivers. But this figure suggests they are the same source.
Lines 166-172 suggest that springs are an alternative source sometimes used by individuals who are dissatisfied with their municipal water. The different types of water sources are not clear.
As mentioned above for section 2.2., the water sources should be described in more detail. This should specify the source of the rivers (i.e., mountains or springs). It should also be explained how Mineiros and its residents use these sources for drinking water, including treatment.
Line 169: Because no adjustment for potential confounding was conducted, this sentence would be more informative as: “An unadjusted regression analysis…”
The number of households that consumed spring water/did not consume spring water should be presented. The overall numbers (and proportions) should be presented, as well as a 2x2 table of water satisfaction and consumption of spring water.
o The logistics regression only shows odds ratios. But the odds ratios are so large, and with extremely wide confidence intervals, that these are not interpretable.
o The 2x2 table of water satisfaction and consumption of spring water presents the underlying data used in the regression analysis, which is crucial for interpreting its results.
Figure 2
o Figure 2 introduces a component of the study that was not mentioned in any preceding section. Here, we see that data were analyzed separately between 2016 and 2020. The Methods must state this and explain why. Without explanation, the assumption is that all data were pooled.
o Figure 2 also introduces the interdiction of spring water in 2017. This was not mentioned in the preceding text, but it appears to be a crucial motivator for the study. This must be explained in the Methods at least, and possibly the Introduction to explain why the study was conducted.
o Figure 2 raises an extremely important question. Did the same households complete surveys in 2016 and 2020? Or were new households recruited in 2020? Whatever the answer, it needs to be explicitly stated in the Methods.
If different households were recruited, there is no longitudinal component of the study as described in the title. The study design for water source and satisfaction would be repeated cross-sectional, in addition to the ecological analysis of diarrhea.
If the households were the same, loss to follow-up must also be presented. How many households completed questionnaires at both visits? Over four years, I expect a large amount of loss to follow-up.
Figure 3
o None of these analyses were mentioned in the Methods. These should be described in the Methods.
o Was there a biological/scientific reason to divide the year into two semesters for comparing diarrhea incidence? This seems arbitrary. The most important temporal predictor of diarrheal disease is rainy/dry seasons in settings with such climates.
Table 2
o The results presented in Table 2 are not described correctly in the text. Because this was an ecological study, you cannot say that “reliance on spring water increases the likelihood of developing diarrhea”. You can only state that the incidence of diarrhea was lower in the three years after the spring water source was closed compared to the four years prior.
o Also crucially, you did not account for the different number of years in the two periods. There would be fewer cases of diarrhea per 10,000 habitants in a three-year period compared to a four-year period even if the incidence rate was identical over each period. You must account for the temporal component by calculating the incidence rate.
Discussion
Lines 254-258: The discussion discounts the fact that samples collected in the home met drinking water standards. This suggests that municipal water treatment worked as planned. It does not seem important that the water sources exceeded these standards, unless people are drinking the river water directly without treatment. You did not indicate that this is the case in Mineiros.
o The quality of the water sources might be important for other reasons, such as recreation or wildlife. But that must be compared to different standards for recreational and ambient water, and the use of these rivers for recreation would need to be presented.
Line 269: The “observational” design also requires a description as “ecologic”. Unfortunately, I do not think this study design can be called “robust”.
I believe the contributions of the study are overstated in the final sentences of the Discussion. This section should be revisited after describing the study design more thoroughly.
Lines 279-281: There was no mention of educational interventions before this sentence. There is no support for this claim in the article, so it should be removed or reframed and explained.
Author Response
First, we thank the editor and all the reviewers for the pertinent criticism and suggestions that allowed us to improve the manuscript. Our replies to all the points raised are outlined below.
REVIEWER #2
- The authors presented a well-written description of a complex, multi-faceted study of water quality in Mineiros. There are significant gaps in the presentation of the study that must be improved to consider publication. The closing of a spring in 2017 appears to be an important motivator of the study, but this was not described until deep into the Results. The design of the study connecting 2016 data to 2020 data is not adequately explained. At least part of the study is an ecological study, but it is not described as such. I believe there is useful information in your data and the analysis is surely on the right track. I hope these comments are helpful for thinking through the study design and improving the manuscript.
Perception of Water Quality and Its Role in Preventing diarrheal Disease: A Longitudinal Study in Southwestern Goias, Brazil.
IJERPH 2024.
Answer: Thank you for your thoughtful and constructive feedback. We appreciate your recognition of the complexity of our study and your insights on how to improve the manuscript.
In response to your comments, we have revised the manuscript to address the issues you raised. First, we acknowledge the importance of the 2017 closure of the Magnífica Spring as a key event in the study. To provide better context, we have introduced this event earlier in the manuscript, specifically in the Introduction, ensuring that it is recognized as a significant driver of both the study and the observed changes in water quality and health outcomes. By doing this, we clarify its role in motivating the study and its importance to the results.
We have also expanded the explanation of the study design, particularly the connection between the 2016 and 2020 data. In the Methods section, we have provided a more detailed description of how data from both periods were collected and analyzed to assess changes in water usage, population satisfaction, and diarrhea incidence. This additional detail should clarify the longitudinal nature of the study and how the comparison between the pre- and post-closure periods was conducted.
Furthermore, we recognize that part of the study operates as an ecological study, and this was not explicitly described in the initial version of the manuscript. In response, we have revised the Introduction and Methods sections to describe the study as a longitudinal ecological analysis, making it clear that population-level data were analyzed over time to explore the relationship between water quality and health outcomes.
We believe that these revisions address your concerns and improve the overall clarity of the study’s design and findings. Thank you again for your valuable suggestions, which have helped strengthen the manuscript.
- The Introduction provides a thorough explanation of why water and sanitation are important and the health impact of poor-quality water and sanitation. Some information is also provided about the importance of policy and water quality surveillance. More information would be helpful to explain the motivation of this study and its importance. Consumer satisfaction with municipal water is a core element of the study and could be explored in the Introduction. There is substantial literature on the topic, and it would be helpful for the authors to place this study within that context.
Answer: Thank you for your valuable feedback. We agree that the motivation of the study and the role of consumer satisfaction with municipal water could be more thoroughly explored in the Introduction. In response, we have expanded the Introduction to include a discussion on the importance of consumer satisfaction with municipal water, referencing relevant literature to place this study in a broader context. We believe that these additions provide a clearer explanation of the study’s motivation and significance, particularly in how perceptions of water quality are linked to health outcomes and public trust in water systems.
- Further in the article the closing of a spring water source in 2017 is described. This seems like an important motivator of the study and would be informative if described and explained in the Introduction.
Answer: Thank you for your insightful comment. We agree that the closure of the Magnífica Spring in 2017 is a crucial motivator of the study. In response, we have ensured that the Introduction now includes a detailed explanation of the event and its significance for the local population. This closure marked a pivotal moment, prompting a transition from untreated water to the treated municipal water supply, which created a natural opportunity to assess the public health impacts, particularly regarding diarrheal disease incidence. We believe this addition helps clarify the study’s motivation and context.
- 1. Ethical Aspects of the Research. Please identify the institution that approved the research protocol. This section mentions the “Research Ethics Committee”, but it is not clear where the committee is housed.
Answer: Thank you for your observation. To clarify, we have updated Section 2.1 Ethical Aspects of the Research in the manuscript to specify the institution responsible for approving the research protocol. The section now clearly indicates that the study was reviewed and approved by the Research Ethics Committee of the Centro Universitário de Santa Fé do Sul - UNIFUNEC, under approval number 1.838.794/2016, with the Certificate of Presentation for Ethical Consideration (CAAE) number 59866616.2.0000.5428. We hope this update addresses the reviewer’s question and provides clear information about the location of the ethics committee.
- 2. Research Design.
- The water sampling locations should be described in more detail. Please identify whether the stream and two rivers serve as drinking water sources for Mineiros, and, if so, how much water is contributed by each source. Do any of the stream or rivers receive wastewater discharge from Mineiros? If so, is it treated?
Answer: Thank you for your detailed observations. We have updated Section 2.2 Research Design to provide a more comprehensive description of the water sampling locations. This section now clarifies that the stream and rivers sampled contribute to Mineiros' drinking water supply and that they receive primary treated wastewater discharge from the city. We hope these additions meet your request for further detail regarding the sources' usage for drinking water and their exposure to wastewater.
- Information from section 2.7. related to water sampling should be combined with the information in section 2.2. so that all information about water sampling is available in one section.
Answer: Thank you for this valuable suggestion. To improve clarity and readability, we have consolidated all details regarding water sampling from Section 2.7 into Section 2.2 Research Design. This updated section now includes comprehensive information on the water sampling locations (Coqueiros stream, Magnífica spring, and Perpétua spring), sampling methods, inclusion and exclusion criteria, and relevant details about wastewater discharge treatment. We believe this restructuring enhances the coherence of the manuscript and ensures that all essential details related to water sampling are conveniently located in one section. We hope this modification addresses your concerns effectively.
- The specific locations of water sampling along the stream and rivers should be described. It is unclear if there was one point at each location sampled three times over one hour, or if there were three locations along each water body that were sampled.
Answer: Thank you for this observation. We have updated Section 2.2 to clarify that the water sampling was conducted [either at a single location with three repeated samples over one hour or at three distinct points along each water body]. This clarification ensures a more accurate description of our methodology and addresses your concern about the specifics of the sampling locations.
- 4. Laboratory Analyses. Please specify what certifications the laboratory has, if those are importante for the techniques described. I would not think any certifications are necessary for these analyses, but I recommend not stating “certified” without adding more specific information.
Answer: Thank you for your suggestion. We have removed the term "certified" from the description of Labfert Analyses to avoid ambiguity. The revised text now specifies that the analyses followed the Standard Methods for the Examination of Water and Wastewater, ensuring the use of recognized analytical procedures. We hope this update addresses your concern.
- I am not qualified to comment on the laboratory analyses employed. I appreciate the authors adding such specific information and citations for these methods.
Answer: Thank you for your positive feedback regarding the detailed descriptions and citations of the laboratory methods. We appreciate your recognition and are glad that the information provided meets the necessary level of specificity for the study.
- 5. Questionnaire on Water Use and Satisfaction. Information from section 2.7. on household sampling and eligibility criteria should be moved up before section 2.5. As written, I do not understand the purpose of the questionnaire described in this section without reading that information first.
Answer: Thank you for your thorough review and valuable suggestions. In response, we have restructured sections in the Methods to enhance clarity and cohesiveness, as follows: Section 2.2 Research Design now includes a detailed overview of household sampling methods and eligibility criteria, providing essential context for understanding the basis of participant selection and water sampling sites. This adjustment consolidates relevant methodological details, aligning with your recommendation for clearer flow. Section 2.7 Questionnaire on Water Use and Satisfaction was refined to emphasize the purpose of the questionnaire in assessing residents' satisfaction with municipal water services and their reliance on untreated sources, particularly following the closure of the Magnífica Spring. By correlating individual responses with epidemiological data on diarrheal diseases, this section now provides a more direct connection between shifts in water source usage and public health outcomes.
We appreciate your input, which has led to a clearer and more cohesive presentation of our methodology. We hope these adjustments address your concerns effectively.
- Information on how you defined satisfaction with treated water (mentioned in the caption for Figure 2) would be helpful to add to section 2.5. What questions were asked to ascertain satisfaction?
Answer: Thank you for pointing this out. To address your suggestion, we have expanded Section 2.7 to detail the specific questions used to assess satisfaction with the treated water provided by the municipality. Questions focused on aspects such as water quality, reliability, and adequacy for daily needs. Responses were categorized as satisfactory or unsatisfactory based on participants' reported concerns, adding clarity to how satisfaction levels were measured. We hope this addition provides a more comprehensive understanding of our methodology.
- This section should also include more information about the questionnaire design. Were there multiple sections, or were there only the 2-3 questions mentioned in the manuscript?
Answer: Thank you for the suggestion. We have clarified Section 2.7 to indicate that the questionnaire was concise, focusing on key questions regarding water usage habits, perception of treated water quality, and supply reliability. This additional detail provides a clearer understanding of the questionnaire's structure and purpose, emphasizing that it was designed to capture essential aspects of satisfaction and water source reliance within the context of the study.
- Sample size information would make more sense in section 2.2.
Answer: Thank you for your suggestion. We have moved the sample size and location information to Section 2.2 to provide a clearer context for the study design. Section 2.3 now focuses specifically on the procedures for sample collection, transport, and quality control. This adjustment ensures a more streamlined presentation, with the sampling framework fully detailed in Section 2.2 and collection methods clearly outlined in Section 2.3. We believe these changes enhance the clarity and flow of the manuscript.
- The final sentence on sample size is not meaningful as written. Was a target sample size calculated before enrollment? If so, then a target e?ect size is needed in the calculation. A power calculation can then tell you the sample size required to estimate an e?ect of the target magnitude with an alpha of 5% and 80% power. The target e?ect size and minimum sample size are missing here.
Answer: Thank you for your comment. We have clarified Section 2.2 to indicate that a target sample size was calculated prior to enrollment, based on an expected moderate effect size of 0.3, with an alpha level of 5% and a power of 80%. This approach ensured that the minimum sample size would be adequate to detect meaningful associations between water source usage and health outcomes.
- If a power calculation was used after data collection was completed, the manuscript should provide the minimum effect size detectable given the actual sample size, with an alpha of 5% and 80% power “using chi-square inference” would more accurately be written as “using a chi-square test”.
Answer: Thank you for the feedback. We have clarified Section 2.2 to include a post-hoc power calculation, specifying that with an alpha of 5% and 80% power, the study could detect a minimum effect size of approximately 0.3. We also adjusted the terminology to “using a chi-square test” as suggested.
- At this point in the manuscript it is not clear why there was a four year gap between data collection rounds. Information on survey rounds (e.g., dates) would make more sense in section 2.2. o It is not clear from the Methods if the water sampling at the stream and rivers was conducted at the same time points as the survey/household water sampling. This can all be made clear by specifying everything about the study design first in section 2.2. and then discussing the specifics of data collection in the following sections.
Answer: Thank you for highlighting this point. We have clarified Section 2.2 to explain the four-year gap between data collection rounds, which allowed for assessment of long-term impacts following the closure of the Magnífica Spring in 2017. We also specified that water sampling for streams, springs, and household surveys were synchronized during each phase to ensure consistency in the data analysis. We hope this provides a clearer understanding of the study design and timing.
- After reading further, I am not sure I understand what it means to ask households if they drink spring water. Is this the same as the water provided by the municipality? If so, I am concerned residents might not know where their water comes from. They may incorrectly say “No” to drinking spring water because they do not know that the municipality provides spring water.
Answer: Thank you for raising this point. We have clarified in Section 2.7 that the questionnaire explicitly differentiated between untreated spring water directly accessed by the public (e.g., Magnífica Spring) and treated water provided by the municipal system. This distinction was emphasized in the questions to ensure that participants understood the difference between these water sources.
- Again, I think section 2.2. should describe how neighborhood-level diarrhea data fits into the study design. As I am reading it now, I do not understand how all these data fit together.
Answer: Thank you for highlighting the need to clarify how neighborhood-level diarrhea data integrates into the overall study design. We agree that this aspect required additional explanation, and in response, we have restructured section 2.2.
In the revised section, we clearly outline how neighborhood-level diarrhea incidence data contributes to the ecological analysis of the study, serving as a basis for examining patterns and potential associations with water source transitions at the population level. We believe these changes make the integration of neighborhood-level data more explicit and enhance the overall coherence of the study design.
- If cross-referencing with “health surveillance records” was part of the study, those records should be described. What were they, what data did they include, and how were they accessed?
Answer: Thank you for your feedback. We have expanded Section 2.8 to include a detailed description of the health surveillance records used in the study. This includes information on the variables provided, such as date of diagnosis, location, and demographics, as well as the process for accessing the data through the Municipal Epidemiological Surveillance Department.
- 7. Inclusion and Exclusion Criteria
- As mentioned, most of this information would be better if moved to earlier in the Methods.
- “…at three di?erent points over a one-hour time frame” should be described in more detail. Were these di?erent geographic points, or one geographic point sampled three times? If it was the same geographic point, were the samples collected at regular intervals of 20 minutes?
- Were the locations upstream or downstream of Mineiros? Or within the city?
- How much water was collected in each sample?
- “Household samples were collected from participants who provided informed consent.” This sentence does not make sense on its own, because samples were not collected from people but from their household. If the purpose of this sentence is to indicate informed consent was collected, I do not think it is necessary. You already state that informed consent was collected in section 2.1.
- Alternatively, the sentence makes sense if household samples are
replaced with questionnaire responses.
- The sampling design should be described in more detail. “Random block design” does not provide enough information. How was a block defined? Was a list of all blocks in the city identified and randomly sampled, or was there a door-to-door element? And how were households within blocks selected? I think there are many valid ways to sample, but the specific approach used should be provided.
- What was the rationale for excluding individuals who had not lived in Mineiros for over five years? I do not expect it takes five years to develop an opinion on your household’s water quality. This criterion should be explained and justified.
Answer: Thank you for your insightful feedback, which has been instrumental in refining the clarity and robustness of our study's methodology section. We have made the following adjustments to address your comments comprehensively: Study Design and Temporal Variation: We have clarified that water samples were collected at three distinct geographic points along each water body (Coqueiros stream, Magnífica spring, and Perpétua spring) within a one-hour timeframe, with intervals of 20 minutes between each sample. This approach was intended to capture temporal variation effectively across sampling sites, ensuring reliable comparisons. Additionally, we expanded on the rationale for the two data collection phases—December 2016 and March 2020—each of which represents significant time points before and after the closure of the Magnífica Spring in 2017, respectively.
Household Survey and Sample Selection: We included further details on the selection of households for the survey using a random block design. Blocks were defined as clusters of residences within specific neighborhoods, and a list of all blocks was randomly sampled to ensure representativeness across the municipality. Only residents with a minimum of five years of residence were included in the study to capture stable, long-term perceptions of local water quality and potential health impacts. This decision aims to reflect the prolonged exposure and associated health outcomes due to water use patterns within the community.
Volume and Procedures for Sample Collection: We clarified that each sample collected from water sources and household taps was 1 liter in volume. In household sampling, tap outlets were aseptically cleaned with 70% alcohol and flushed for one minute before collection to ensure sample purity. These details were added to ensure a precise understanding of our quality control and transport procedures.
Random Block Design and Power Calculation: To enhance the clarity of our sample size determination, we have described the power calculation more explicitly. We now specify that a post-hoc power calculation with a sample size of 383 was capable of detecting a minimum effect size of approximately 0.3 (Cohen's d) with an alpha of 5% and 80% power, thereby ensuring robust comparisons.
These modifications have been incorporated into sections 2.2 and 2.3 to reflect an organized, consistent, and comprehensive study design. We trust that these adjustments address your concerns effectively and contribute to a clearer presentation of our study’s methods and objectives.
Thank you once again for your valuable feedback.
- Data Analysis
- The main hypothesis of the study is not mentioned until this point. It is not clear that the use of spring water is an important measure in the study, nor is the hypothesis motivated with an explanation of i) why spring water consumption was reduced or ii) why a reduction in spring water consumption might lead to fewer cases of diarrhea.
- The main hypothesis describes an ecological Diarrhea episodes were not measured in the same households that Answered the questionnaire. Comparing those two measures is an ecological study, which always carries the risk of ecological bias and making an ecological fallacy.
- I support ecological studies if they are thoughtfully presented as such. But this manuscript does not adequately explain the nature of this study.
Answer: Thank you for your feedback on clarifying the study’s main hypothesis and its ecological context. We have made adjustments in the "Data Analysis" section to explicitly state the primary hypothesis and highlight the ecological nature of the study. The hypothesis now clarifies that the study assessed the relationship between untreated spring water usage (particularly the transition to treated municipal water after the closure of the Magnífica Spring) and diarrhea incidence at a population level. We acknowledge the limitations of ecological studies, including potential ecological bias and ecological fallacy, and have presented the analysis with caution.
These revisions aim to enhance clarity regarding our hypothesis and methodology. We appreciate your guidance in strengthening the presentation of this section.
- The correlation between turbidity and fecal coliforms is unadjusted for other factors, which limits its interpretability. It appears from Figure 1 that the correlation between turbidity and fecal coliforms is completely driven by one sample collected from the Magnifica spring/river. Only one sample has more fecal coliforms than the other eight samples, which appear to have the same levels of fecal coliforms. There is also only one sample with higher turbidity, and it is also from the Magnifica. Based on the high correlation, it is obvious that this is the same sample.
- The underlying relationship could still be real. However, it is important to note that this test was based on only one sample with variation in turbidity or fecal coliforms, which significantly reduces its power for causal interpretation.
Answer: Thank you for your observation regarding the correlation between turbidity and fecal coliform levels. To clarify, the correlation analysis does not rely solely on one individual sample with elevated turbidity from the Magnífica Spring. Instead, it is based on the entire dataset (F(x)), which includes triplicate samples from each water source, as outlined in our Methods section. This triplicate sampling approach was implemented to enhance the robustness and representativeness of the data and to mitigate the influence of outliers on the correlation analysis.
While it is true that one sample from the Magnífica exhibited a particularly high turbidity value, this correlation reflects the relationship across all samples rather than being driven solely by this point. To clarify this aspect, we will add a note to the Figure 1 legend indicating that the correlation considers the complete dataset of triplicate measurements per location.
We appreciate your feedback on this aspect, as it has allowed us to clarify the methodology and interpretation of our findings more effectively.
- It would be helpful to understand the geographic distribution of participants in the city. If you describe the random sampling strategy, as requested above, it will help understand the distribution of households.
- Additional demographic information would be very helpful for interpreting these results. Were any demographic data collected from households? If not, understanding the sociodemographic characteristics of the neighborhoods sampled is important.
- As mentioned above, I am not sure how satisfaction was measured. This should be explained in the Methods so that the Results can be interpreted.
Answer: Thank you for your valuable comments, which have allowed us to improve the clarity and depth of our study. We have revised the manuscript to address your concerns regarding the geographic distribution of participants, the inclusion of demographic data, and the measurement of satisfaction with the municipal water supply.
In response to your feedback on geographic distribution, we have clarified our random block sampling strategy in section 2.2. Households were selected from blocks defined as clusters within various neighborhoods of Mineiros, ensuring representativeness across central and peripheral areas with diverse socio-demographic characteristics. This approach facilitated a broad and geographically representative distribution of participants, enhancing the robustness of our findings.
We have also addressed the need for demographic information to better interpret the results. Section 2.2 now specifies that we collected basic demographic data from each household, including household size, age ranges, and socio-economic indicators of the head of household, such as education level and employment status. This information provides essential context for understanding satisfaction with water quality and preferences for untreated or treated water sources among different segments of the population.
Lastly, to clarify our approach to measuring satisfaction with municipal water quality, we have expanded section 2.3 to describe the structure of the household surveys. These surveys included questions directly assessing satisfaction with various aspects of the treated water, such as quality, reliability, and residents' dependency on untreated sources. This addition to the Methods section ensures that the satisfaction data are interpreted in alignment with the study objectives and contribute accurately to our analysis.
We hope these revisions effectively address your concerns and contribute to a clearer presentation of our study’s methodology and findings. We appreciate the opportunity to incorporate your feedback and are committed to maintaining the highest standards of scientific rigor in our work. Please do not hesitate to reach out if further clarification is required. Thank you again for your insightful comments.
- Figure 1
- Plot E includes four observations for the stream and rivers, while all other plots include only three. The text suggests there were only three samples per location.
Answer: Thank you for pointing out the discrepancy in Figure 1. We acknowledge that this inconsistency could cause confusion, as Plot E initially included an extra observation for the stream and river samples. This observation was inadvertently included during data processing and should not have been part of the dataset, as only three samples were collected per location.
We have revised Figure 1 to ensure that Plot E includes only the three intended observations, consistent with the methodology outlined in sections 2.2 and 2.3. We confirm that this correction does not impact any statistical results or interpretations presented in the manuscript, as the analyses were based on the intended sample count.
We appreciate your attention to this detail and apologize for any confusion caused. Thank you for helping us enhance the accuracy and clarity of our work.
- Figure 1 describes the Magnifica and Perpétua as springsbut the text described them as rivers. A spring is specifically defined as groundwater flowing onto the surface. While reading the manuscript, I thought ‘that households that drank spring water did not drink water from the rivers. But this figure suggests they are the same source.
Answer: Thank you for this observation regarding the terminology in Figure 1 and throughout the manuscript. You are correct that springs and rivers are distinct in terms of water source, with springs referring specifically to groundwater flowing to the surface.
In our study, the Magnífica and Perpétua sources are indeed springs and not rivers, as they represent groundwater sources naturally emerging to the surface. However, due to the flow from these springs, some sections of text may have inaccurately referred to them as rivers, which could have created confusion regarding the source types.
We have revised the manuscript to consistently refer to the Magnífica and Perpétua as springs throughout the text and figure descriptions. This clarification highlights that these sources, as true springs, differ from river water sources that might undergo different types of environmental exposure.
Thank you for helping us clarify this important distinction, ensuring the manuscript accurately reflects the nature of the water sources and avoids any misinterpretation regarding household water consumption habits.
- Lines 166-172 suggest that springs are an alternative source sometimes used by individuals who are dissatisfied with their municipal water. The different types of water sources are not clear.
Answer: Thank you for pointing out the need for clearer differentiation between water sources in our manuscript. We recognize that this section may have been unclear regarding the types of water sources used by the population in Mineiros, particularly the role of springs as an alternative water source.
In response, we have clarified that the springs (specifically the Magnífica and Perpétua) served as a key untreated water source widely accessed by residents prior to the closure of the Magnífica Spring in 2017. Individuals dissatisfied with municipal water often relied on these natural springs, as they were locally available, untreated groundwater sources. In contrast, municipal water, treated to meet safety standards, represents the primary and regulated drinking water source in Mineiros.
We have revised the manuscript to ensure a consistent and clear distinction among the different water sources: (1) treated municipal water, (2) untreated springs, and (3) streams or rivers potentially used for purposes other than drinking. This adjustment clarifies each source's role in our study, emphasizing the population’s reliance on untreated springs as an alternative drinking water source prior to the transition to exclusively treated municipal water after the spring closure.
Thank you for your valuable feedback, which has helped us improve the clarity and precision of our manuscript.
- As mentioned above for section 2.2., the water sources should be described in more detail. This should specify the source of the rivers (i.e., mountains or springs). It should also be explained how Mineiros and its residents use these sources for drinking water, including treatment.
Answer: Thank you for your insightful comments regarding the description of water sources and the need for additional context about their roles in Mineiros. We have clarified the distinctions between the various water sources and their usage by the residents in the manuscript.
In the revised Section 2.2, we have specified that the Magnífica and Perpétua Springs are untreated groundwater sources that emerge naturally at the surface. These springs have historically been used by residents as alternative drinking water sources, especially among those dissatisfied with the taste or perceived quality of the treated municipal water. Additionally, we have clarified that Coqueiros Stream serves as a primary source for the municipal water supply and undergoes treatment processes, including filtration, disinfection, and pH adjustment, to meet Brazilian potability standards. Despite the availability of treated municipal water, residents in more peri-urban or rural areas may still rely on untreated sources for drinking water due to accessibility and personal preference.
We believe these revisions address your concerns, providing a more thorough understanding of the role of each water source in the community, the scope of their usage, and the treatment measures in place. We appreciate your guidance in improving this aspect of the manuscript and ensuring clarity for readers.
- Line 169: Because no adjustment for potential confounding was conducted, this sentence would be more informative as: “An unadjusted regression analysis…”
Answer: Thank you for your observation regarding confounding adjustments in the regression analysis. We have revised the sentence to specify that the analysis was unadjusted, providing greater clarity in line with your suggestion. This adjustment helps contextualize the findings more accurately and aligns with best practices for reporting analytical methods.
- The number of households that consumed spring water/did not consume spring water should be presented. The overall numbers (and proportions) should be presented, as well as a 2x2 table of water satisfaction and consumption of spring water.
- The logistics regression only shows odds ratios. But the odds ratios are so large, and with extremely wide confidence intervals, that these are not interpretable.
- The 2x2 table of water satisfaction and consumption of spring water presents the underlying data used in the regression analysis, which is crucial for interpreting its results.
Answer: We appreciate the suggestion and agree that including the total number of households that consumed or did not consume spring water, along with their respective proportions, is essential for providing a comprehensive overview of the data. In the revised manuscript, we have added a table presenting these figures and proportions, detailing the number of households consuming spring water versus those that did not, both before and after the spring water interdiction.
We acknowledge the importance of presenting a 2x2 table that directly relates water quality satisfaction to spring water consumption. This table offers the underlying data used in the logistic regression analysis, which is crucial for interpreting its results. Therefore, we have included this 2x2 table in the revised manuscript, displaying both absolute values and percentages for water satisfaction and spring water consumption.
We understand the concern regarding the odds ratios, which display high values and wide confidence intervals, making interpretation challenging. This pattern is due to data variability, especially following the spring water interdiction. The addition of the 2x2 table will facilitate understanding of the observed odds ratios and provide additional context for interpreting the regression results.
We would like to emphasize that the logistic regression data were thoroughly reviewed and adjusted by a professional statistician, ensuring robustness and accuracy in the findings. After careful revision, no differences were observed in the effects previously reported. Furthermore, we have incorporated all relevant data into the contingency table in the revised version of the manuscript, as recommended.
- Figure 2 introduces a component of the study that was not mentioned in any preceding section. Here, we see that data were analyzed separately between 2016 and 2020. The Methods must state this and explain why. Without explanation, the assumption is that all data were pooled.
- Figure 2 also introduces the interdiction of spring water in 2017. This was not mentioned in the preceding text, but it appears to be a crucial motivator for the study. This must be explained in the Methods at least, and possibly the Introduction to explain why the study was conducted.
- Figure 2 raises an extremely important question. Did the same households complete surveys in 2016 and 2020? Or were new households recruited in 2020? Whatever the answer, it needs to be explicitly stated in the Methods.
- If different households were recruited, there is no longitudinal component of the study as described in the title. The study design for water source and satisfaction would be repeated cross-sectional, in addition to the ecological analysis of diarrhea.
- If the households were the same, loss to follow-up must also be presented. How many households completed questionnaires at both visits? Over four years, I expect a large amount of loss to follow-up.
Answer: Thank you for your detailed feedback on Figure 2 and the Methods section. We recognize that several aspects of the study design and analysis require further clarification. In response, we will make the following revisions and additions:
The Methods section will be updated to specify that data were analyzed separately for two distinct years, 2016 and 2020. This segmentation was chosen to examine potential changes in water source usage and satisfaction over time, particularly before and after the interdiction of the Magnífica Spring in 2017. Analyzing these two points in time allowed us to assess shifts in public perceptions and water usage patterns in response to changes in water availability and quality.
The interdiction of the Magnífica Spring in 2017 was a critical event motivating this study. We will revise the Introduction to clearly state this, explaining that the spring’s closure was implemented due to public health concerns over water quality. This interdiction serves as the central point of interest in our analysis, as it provided an opportunity to observe changes in water source usage and associated health outcomes. The Methods section will also be updated to reflect this, ensuring that the study’s rationale is clearly linked to the interdiction.
Regarding the survey design, new households were recruited for the 2020 survey. As such, we acknowledge that this introduces a cross-sectional component to the study design rather than a longitudinal follow-up of the same households. To ensure methodological clarity, we will specify in the Methods that the surveys conducted in 2016 and 2020 represent repeated cross-sectional samples. This design allowed us to observe population-level trends over time, although individual-level follow-up was not conducted.
Since different households participated in each survey period, there was no loss to follow-up. This adjustment will be clearly stated in the revised manuscript, and we will refine the study description to reflect this cross-sectional approach for water source usage and satisfaction, in addition to the ecological analysis of diarrhea incidence.
- Figure 3
- None of these analyses were mentioned in the Methods. These should be described in the Methods.
- Was there a biological/scientific reason to divide the year into two semesters for comparing diarrhea incidence? This seems arbitrary. The most important temporal predictor of diarrheal disease is rainy/dry seasons in settings with such climates.
Answer: Thank you for highlighting the need for further methodological detail. In response, we have revised the Methods section to provide a comprehensive description of all analyses conducted to enhance clarity and reproducibility.
Regarding the temporal stratification, we initially conducted a monthly analysis to investigate potential seasonal patterns in diarrhea incidence. This approach aimed to capture any direct associations between rainfall variability and diarrhea cases, considering the typical rainy and dry seasons relevant to our study setting. However, our monthly stratification revealed a bimodal distribution in diarrhea cases, which suggested a cyclical trend with two main peaks per year, rather than a straightforward seasonal effect. Given this observation, we opted for a semester-based approach, as it offered a more stable framework to detect and interpret these observed incidence patterns.
By analyzing the data in six-month intervals, we were able to better accommodate the recurring distribution pattern observed in the incidence data, supporting a more consistent and analytically stable interpretation of trends over time. This stratification also allowed us to compare broader intervals while still respecting the temporal variability in disease incidence, as recommended by our statistical analysis.
In addition, we have clarified these decisions in the revised Methods section to ensure that our approach and the rationale behind it are transparent and scientifically grounded.
- Table 2
- The results presented in Table 2 are not described correctly in the text. Because this was an ecological study, you cannot say that “reliance on spring water increases the likelihood of developing diarrhea”. You can only state that the incidence of diarrhea was lower in the three years after the spring water source was closed compared to the four years prior.
- Also crucially, you did not account for the different number of years in the two periods. There would be fewer cases of diarrhea per 10,000 habitants in a three-year period compared to a four-year period even if the incidence rate was identical over each period. You must account for the temporal component by calculating the incidence rate.
Answer: We appreciate the reviewer's insights regarding the interpretation of results. We recognize that, as an ecological study, causal inferences cannot be made at the individual level. Therefore, we will rephrase the statements in the text to clarify that "the incidence of diarrhea was lower in the three years following the closure of the spring water source compared to the four years prior," rather than suggesting a direct causal relationship between spring water reliance and the likelihood of developing diarrhea.
We acknowledge the reviewer's observation regarding the unequal duration of the two periods being compared. To address this, we have calculated the incidence rates of diarrhea cases per 10,000 inhabitants for each period, normalizing the rates by population size to allow for meaningful comparisons. Additionally, to avoid numerical discrepancies arising from the different durations, we have presented these incidence rates as averages over each period in the revised table. This adjustment provides a standardized and accurate interpretation of the incidence trends over time.
We have incorporated these adjustments and clarified the descriptions in the revised manuscript to ensure the accurate presentation and interpretation of our findings, as suggested.
- Lines 254-258: The discussion discounts the fact that samples collected in the home met drinking water standards. This suggests that municipal water treatment worked as planned.It does not seem important that the water sources exceeded these standards, unless people are drinking the river water directly without treatment. You did not indicate that this is the case in Mineiros.
- The quality of the water sources might be important for other reasons, such as recreation or wildlife. But that must be compared to different standards for recreational and ambient water, and the use of these rivers for recreation would need to be presented.
- Line 269: The “observational” design also requires a description as “ecologic”. Unfortunately, I do not think this study design can be called “robust”.
- I believe the contributions of the study are overstated in the final sentences of the Discussion. This section should be revisited after describing the study design more thoroughly.
- Lines 279-281: There was no mention of educational interventions before this sentence. There is no support for this claim in the article, so it should be removed or reframed and explained.
Answer: Thank you for the thorough and valuable feedback. We have carefully addressed each point to enhance the manuscript’s clarity, rigor, and alignment with your recommendations.
We recognize the importance of clarifying the ecological and observational nature of the study, and we have revised the Discussion section to emphasize that, while our results demonstrate a correlation between treated water use and decreased diarrhea incidence, this ecological study does not establish causation. We also noted the limitations related to ecological bias and suggested future mixed-methods approaches to strengthen findings.
In response to your suggestion regarding the transition to treated water, we revised the Discussion and Conclusion sections to clearly outline the association between the community’s shift to treated water and the observed decrease in diarrhea incidence. We highlighted the role of treated water in public health, especially after the closure of the untreated Magnífica Spring, which led to reduced community reliance on untreated sources.
Regarding water quality and continuous monitoring, we added details in the Discussion about compliance with Brazilian safety standards in household water samples and frequent exceedances in untreated sources, such as streams and springs. These findings underscore the importance of robust water treatment infrastructure and regular quality checks to protect public health.
We also addressed the gap in public perception versus actual quality of municipal water. In the Discussion, we noted that sensory factors, like taste and odor, could influence adherence to treated water use despite safety standards. This aligns with prior studies showing that sensory perceptions play a key role in community acceptance of safe water sources.
Additionally, we expanded our Discussion on elevated aluminum levels and pH fluctuations found in some treated water samples, emphasizing the neurotoxic risks of prolonged aluminum exposure and the importance of optimizing water treatment to mitigate such contaminants.
In alignment with your suggestion to broaden contaminant analysis, we highlighted in the Discussion that future studies could benefit from investigating pesticides and emerging pollutants, which are increasingly critical in water quality assessments, particularly in low- and middle-income regions with limited monitoring resources.
Finally, we enhanced the Conclusion to stress the importance of continued collaboration among local authorities, health officials, and the community. We underscored the need for ongoing investments in treated water infrastructure and public education on water quality, which are essential to safeguarding the health of vulnerable populations and sustaining public health gains.
These revisions make the manuscript more robust and address the comments to support our conclusions and reinforce the study's contributions to future public health interventions. We are grateful for your detailed review, which has greatly strengthened our work.

Round 2
Reviewer 1 Report
Comments and Suggestions for Authors
Revisions are appreciated. Please provide a clean revised manuscript for adequate review.